# Changes in the structure and composition of the 'Mexical' scrubland bee community along an elevational gradient

Sergio Osorio-Canadas[1]*, Noé Flores-Hernández[1,2], Tania Sánchez-Ortiz[1], Alfonso Valiente-Banuet[1]

1 Departamento de Ecología de la Biodiversidad, Instituto de Ecología, Universidad Nacional Autónoma de México, Mexico City, Mexico, 2 Departamento de Ciencias Ambientales, División de Ciencias Biológicas y de la Salud, Universidad Autónoma Metropolitana-Unidad Lerma, Lerma de Villada, Mexico

* s.osorio.canadas@gmail.com

**Data Availability Statement:** All relevant data are within the manuscript and its Supporting Information files.

## Abstract

'Mexical' scrubland is a sclerophyllous evergreen Mediterranean-like vegetation occurring in the leeward slopes of the main Mexican mountain ranges, under tropical climate. This biome occupies an elevational range approximately from 1900 to 2600 meters above sea level, which frequently is the upper-most part of the mountains range. This puts it at risk of extinction in a scenario of global warming in which an upward retraction of this type of vegetation is expected. The Mexical remains one of the least studied ecosystems in Mexico. For instance, nothing is known about pollinator fauna of this vegetation. Our main objective is to make a first insight into the taxonomic identity of the bee fauna that inhabits this biome, and to study how it is distributed along the elevational gradient that it occupies. Our results highlight that elevation gradient negatively affects bee species richness and that this relationship is strongly mediated by temperature. Bee abundance had no significant pattern along elevational gradient, but shows a significant relationship with flower density. Interestingly, and contrary to previous works, we obtained a different pattern for bee richness and bee abundance. Bee community composition changed strongly along elevation gradient, mainly in relation to temperature and flower density. In a global warming scenario, as temperatures increases, species with cold preferences, occupying the highest part of the elevation gradient, are likely to suffer negative consequences (even extinction risk), if they are not flexible enough to adjust their physiology and/or some life-story traits to warmer conditions. Species occupying mid and lower elevations are likely to extend their range of elevational distribution towards higher ranges. This will foreseeably cause a new composition of species and a new scenario of interactions, the adjustment of which still leaves many unknowns to solve.

## Introduction

Paleontological evidence indicates that evergreen-sclerophylous Mediterranean-like vegetation originally existed in a belt around North America and Eurasia where the climate was wet and

**Funding:** This study was supported by Dirección General de Asuntos del Personal Académico (DGAPA), Universidad Nacional Autónoma de México (UNAM) with the project PAPIIT clave IN-214020, (https://dgapa.unam.mx/index.php/impulso-a-la-investigacion/papiit). This study was also supported by Consejo Técnico de la Investigación Científica (CTIC)-Dirección General de Asuntos del Personal Académico (DGAPA), Universidad Nacional Autónoma de México (UNAM), with one post-doctoral grant (CJIC/CTIC/0980/2019 and CJIC/CTIC/4698/2020) (to SO-C) (https://dgapa.unam.mx/index.php/formacion-academica/posdoc). The funders had no role in study design, data collection and analysis, decision to publish, or preparation of the manuscript.

**Competing interests:** The authors have declared that no competing interests exist.

warm during the mid-Eocene [1, 2]. Currently, relictual patches of this ecosystem type are found in the five Mediterranean areas under Mediterranean climate. We also found it in the form of the sclerophyllous evergreen scrublands occurring in the leeward slopes of the main Mexican mountain ranges under tropical climate (Sierra Madre Oriental, Sierra Madre Occidental, Eje Neovolcánico and Oaxaca mountains), in an elevational range approximately from 1900 to 2600 meters above sea level (m asl). These plant communities grow in semi-arid tropical climates with summer rainfall and are known as the 'Mexical' in contraposition to the 'Chaparral' from California and Baja California, developing under Mediterranean climate [3]. Surprisingly, the Mexical remains one of the least studied ecosystems in Mexico, even though the scant existing evidence indicates that it harbors high levels of biodiversity [3]. In addition, most studies on Tropical mountain systems assume that these are extensions of lowland ecosystems [4]. But in the case of Mexical, this biome has an origin, common genera, and ecological traits (evergreening, sclerophyllous, leaf angle, resprouting ability), that are much closer to other Mediterranean ecosystems than to the low deciduous forest and derived scrublands, located a few meters below in the same mountain, and whose botanical lineage belongs to a different (neotropical) geoflora [1, 2, 5, 6].

As the Mexical occupies relatively high elevations in mountain systems (in some cases the upper-most part of the mountains), climate change is expected to have a strong impact on this ecosystem. Temperature increases are likely to cause the Mexical to retreat to higher elevations which would seriously compromise its very existence. Reduction and/or alteration of the Mexical would threaten plant species characteristic of this ecosystem, as well as their herbivores, including pollinators [7, 8]. Most of the plants of the Mexical show floral traits compatible with the syndrome of Mediterranean systems of the Tertiary [9, 10], and thus are likely to depend on pollinators for fruit and seed set. However, and in contrast to the pollinator fauna of other Mexican ecosystems, such as the semi-arid areas of Mexico valley [11], the lowland deciduous forest of the Jalisco coast [12], and the lowland tropical forest of the Yucatán Peninsula [13], the pollinating fauna of the Mexical is totally unknown.

Pollinator insects in general, and bees in particular, play a key role in the functioning of terrestrial ecosystems. As much as 85% of the Angiosperms (including 75% of human food crops) depend on insect pollination for sexual reproduction [14, 15]. At the same time, bees and other pollinators have experienced important abundance and diversity declines during the last century [16–20]. The drivers of these declines are partially known and include habitat loss and fragmentation, as well as agricultural intensification and, for some species, the arrival of new parasites and pathogens [21]. Climate change is another likely threat to pollinator populations but the information currently available is insufficient to establish whether climate change should be considered an important driver of bee declines [22–25]. Pollinator populations may respond to climate change in different ways. For instance, they may mitigate the effects of increased temperature through phenotypic plasticity and adaptation [26, 27]. On the other hand, they may migrate to new areas tracking favorable weather conditions [8, 28–30]. If none of these mechanisms works, populations will decline, potentially leading to extinction [16]. Migrations tracking weather conditions may occur along latitudinal and elevational gradients [31, 32]. Elevational gradients, in particular, afford an ideal scenario to study the effects of climate change because they provide a pronounced yet gradual change of climatic conditions (especially temperature) across a relatively small geographical area. Even though experimental manipulation and laboratory studies may help understand short-term responses of organisms (and ecosystems), understanding long-term responses (acclimatization, adaptation, species turnover, changes in community structure) is best accomplished through the study of climatic gradients [4]. For this reason, elevational gradients are increasingly being used as study models to predict potential effects of climate change [4, 33, 34].

In this study we apply a standardized sampling method to characterize bee communities along an elevational gradient of Mexical. We have three objectives: 1) To describe the hitherto unknown bee community of the Mexical; 2) To analyze how the structure (richness and abundance) and composition of this community change with elevation, and 3) To test some abiotic (temperature and precipitation) and biotic variables (flower richness and density) that could explain bee elevational variability.

Based on previous studies [35–47], one would expect bee abundance and richness to decrease along the altitudinal gradient. We also expect changes in community composition along our elevational gradient [45, 46, 48, 49]. Regarding the drivers behind these patterns, we expect that decreasing trends of bee richness and abundance would be associated to decreasing temperatures along the elevational gradient [42, 44, 47, 50]. We also expect a positive relationship between bee abundance and flower abundance, as it has been reported along elevational gradients [40, 42, 51, 52], and it is also known from some studies not associated with elevational gradients [53, 54].

## Materials and methods

### Ethics statement

Field work was conducted with the permission of Subsecretaría de Gestión para la Protección Ambiental (SGPA) (permit No. SGPA/DGVS/6790/19), belonging to the Secretaría de Medio Ambiente y Recuros Naturales (SEMARNAT). Our study does not involve any endangered nor protected species according to the NOM-059 of SEMARNAT. The study area is not located in a protected area by the Mexican government.

### Study area

The study area chosen to conduct field work is in the 'Mixteca Alta' region, in north Oaxaca state (Mexico), in the boundaries between Oaxaca and Puebla states (between 16°45′ and 18° 22′ N latitude, and between 96°59′ and 98°27′ W longitude). The climatology of the region includes a rainy season (May-October) and a dry season (November—April). Due to elevation, temperature is relatively smooth or even cold, swinging annually, between 6°C to 28°C (only rarely bellow 2°C or above 32°C). Great part of rainfall occurs along 31 days centered at around September 5th, with a total mean accumulation of 166 mm (although there is also a first peak of rainfall along 31 days centered at around June 28th, with a total mean accumulation of 163 mm) [55]. Evergreen sclerophyllous vegetation ('Mexical') in this zone is located along a belt covering the mountain ranges between Puebla and Oaxaca, between 1850 and 2500 m asl. More specifically, our study sites are located in an area that covers the municipalities of Villa de Tamazulápam del Progreso, Villa de Chilapa de Díaz, and San Andrés Lagunas.

We selected 19 plots of 450 m$^2$ (30 x 15 m) approximately, encompassing an overall area of 280 km$^2$ (Fig 1). Distances between plots ranged from 0.5 to 15.8 km. Plots ranged in elevation from 1850 to 2500 m asl (see S1 Appendix). The 19 selected plots share the same basic vegetation type (Mexical-type scrubland), soil type and recent disturbance history. Plant composition varies locally from plot to plot, but is always largely dominated by basic Mexical elements [3]. Other taxonomic elements could be mixed in the upper and lower parts of our elevational range, as the highest elevation plots are close to the ecotone with pine or oak-type vegetation, and the lower elevation plots are close to the ecotone with lowland deciduous forest.

### Variable elevation

The orography of our study zone displayed a distribution in which elevation changes do not occur along a typical lineal mountain, but more or less irregularly over the territory.

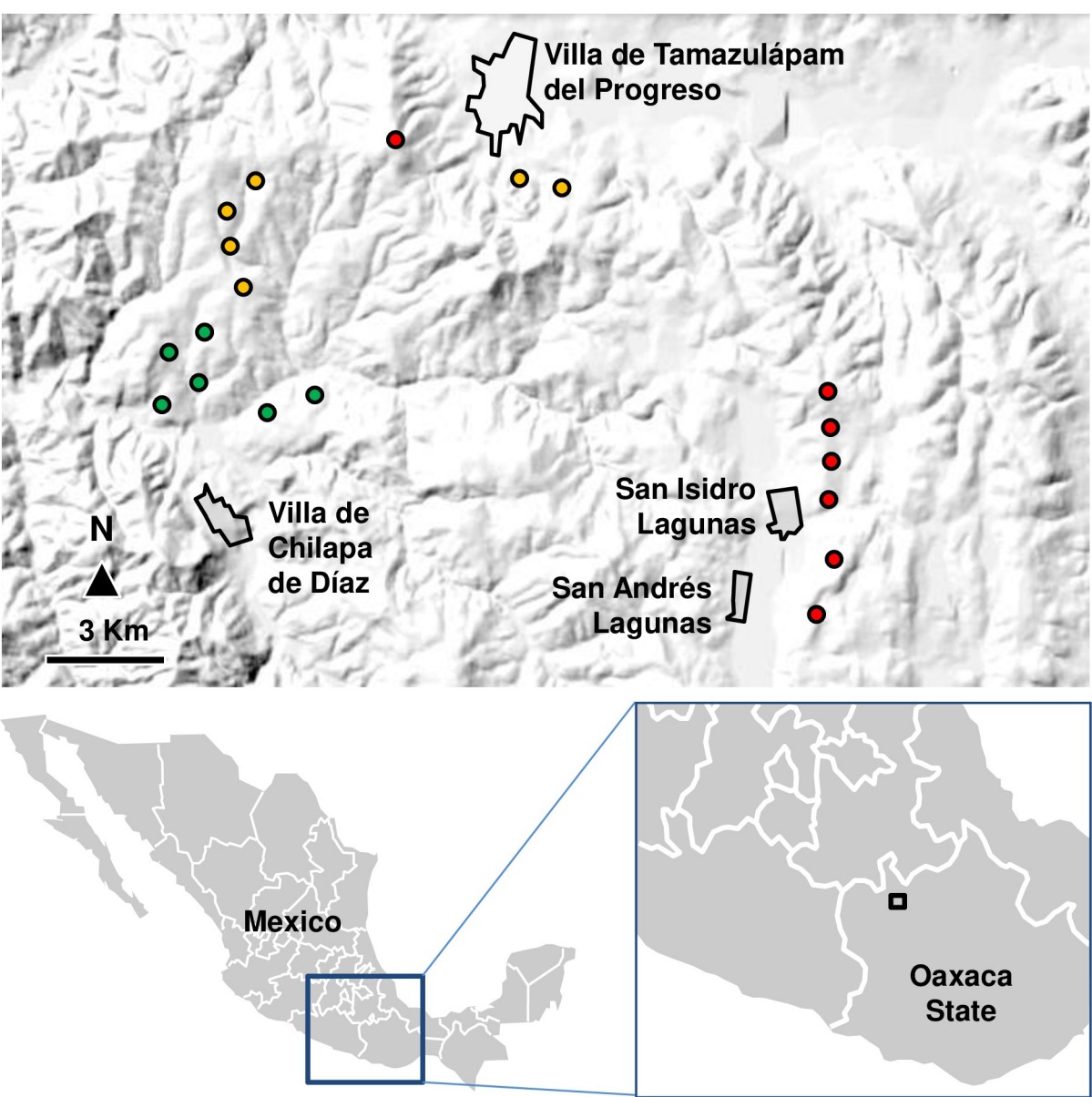

**Fig 1. Study area map.** Location of plots of our survey on an elevation map of the study area. Red dots: plots ranging between 2405 and 2500 meters above sea level (m asl), orange dots: plots ranging between 2110 and 2176 m asl, and green dots: plots ranging between 1850 and 1925 m asl. (When elevation was considered as categorical variable, red dots represented 'high' elevation category plots (~'2450 meters' above sea level (m asl)); orange dots: 'mid' elevation category plots (~'2150 m' asl); green dots: 'low' elevation category plots (~'1850 m' asl). Names of the main municipalities of the study area are shown (San Isidro Lagunas is a village included in San Andrés Lagunas municipality). Black line quadrate insert in Oaxaca State map corresponds approximately to the upper map with relief of the study area.

Specifically, we found Mexical-type scrubland in this area at as low as 1850 m asl (transitioning to low deciduous forest, a little bit lower), and at as high as 2500 m asl (transitioning to pine-wood-type vegetation or oak-type vegetation, depending on the sites, a little bit higher). We studied changes in elevation taking the variable elevation as continuous, although plots were selected relatively grouped in three levels: 6 plots ranging between 1850 and 1925 m asl, 6 plots between 2110 and 2176 m asl, and 7 plots between 2405 and 2500 m asl. The minimum and maximum elevations correspond to those at which we found Mexical-type scrubland. This

yielded a total of 19 plots (see Fig 1). We choose plots with similar conditions of slope and aspect as far as possible.

We also considered variable Elevation as a categorical variable to conduct all analyses described below (see 'Statistical Analysis' section below) for comparison. We established three elevation categories: 'low' (including 6 plots around 1850 m asl), 'mid' (including 6 plots around 2150 m asl) and 'high' (including 7 plots around 2450 m asl). As analyses results were qualitatively equivalent they are not shown in main text, but they are shown in S5 Appendix.

## Climatic variables

Temperature gradient associated to elevational gradients is thought as one of the main drivers behind changes in biotic communities along elevational gradients [34]. Precipitation is another important climatic variable that can affect direct or indirectly bee presence along elevational gradients [56]. We obtained these two variables from corresponding raster layers for GIS of 'Climatic Atlas of Mexico' [57]. We obtained Mean Annual Temperature (˚C) (MAT, henceforth) and Mean Annual Precipitation (mm) (MAP, henceforth) for each plot. These data represent an average from a series of 109 years recorded data for during years 1902 to 2011.

## Bee sampling

We conducted 3 surveys (late September 2019, late October 2019, and late January 2020-early February 2020). In each survey and plot, we placed 6 sampling stations distributed in two parallel rows (3 stations in each of the two rows), with a distance of 15 m between stations in the same row, and with a distance of 15 m between the two rows. Following Westphal et al. [58], each station was composed of 3 pan traps (19-cm-diameter plastic bowls painted yellow, white and blue, respectively, with UV-reflecting paint in the case of yellow and blue, bright in the case of white). Traps were not held in metallic bars as in Westphal el al. [58]. Instead, they were located on the ground, the bowls being ~3 m apart from each other in each station. We searched for clears in the flowering vegetation to set the bowls on the ground. On each sampling day, traps were set on the ground and filled with water containing a small amount of detergent and we took note of the exact hour at which all the bowls were full of soapy water. These bowls were collected at the same hour next day, so that all bowls were active for 24 hours. We were able to complete this process for 6–7 plots on each day. To avoid the influence of weather conditions, surveys were conducted simultaneously in six random plots, two belonging to each one of our three groups of elevation. Then, we were able to survey our 19 plots in 6 consecutive days (one day to set pan-traps in each plot, and the following day to collect insect samples in traps laid the day before). Pan trapping has been shown to underestimate bee species richness compared to netting of flower visiting insects [58]. However, this method avoids collector bias and allows to apply the same sampling effort to each plot, so that samples of all our 19 plots were totally comparable, which was our main concern. Captured specimens were dried and pinned for taxonomic identification in the laboratory. This collection is deposited at the Ecology Institute Entomological Collection, Autonomous National University of Mexico (UNAM), Mexico City, Mexico. From these samples we obtained measures of bee species richness (number of bee species captured), bee abundance (number of bee individuals captured) and bee community composition (abundance of each bee species) for each plot. To obtain the final value for each variable, we lumped together all bee species and all bee individuals sampled per plot for the three surveys. Honey bee (*Apis mellifera*) is known to be a human-managed species in our study area, especially for honey production. As their species distribution could have been modified by human managing, we exclude this species from all analyses. This species was present in all plots.

## Flower resources

We considered flower variables, as principal bee food resource and, in consequence, important co-variables determining presence and abundance of bees [59]. We quantified two variables: flower species richness and flower density in each of our 19 plots. To estimate flower species richness and flower density, we counted all flowers belonging to every flower species along two 20 m$^2$ transects arranged following the two lines in which sampling stations were settled. This was done three times, one for each bee sampling conducted. Some previous studies have shown that pollen and nectar density per plot are highly correlated, and both variables are also correlated to flower density [60]. Therefore, we used flower density as a measure of flower resources in all analyses (number of flowers/m$^2$ in each plot). We also used flower species richness as a variable in our analyses. To obtain the final value for each variable, as in the case of bees, we lumped together all flowering species and all flower individuals sampled per plot for the three surveys.

## Statistical analysis

**Bee community structure, flower community structure and climate vs elevation.** We determined the relationship between bee species richness, bee abundance, flower species richness, flower density, MAT and MAP, each one as a response variable versus the variable elevation (six different models), taking into account a possible variation of these variables in relation to geographic distance (spatial autocorrelation). Previously, we used Moran's I test to explore spatial autocorrelation of each one of this response variables. These analyses were conducted with the statistical package 'ape' [61] for R version 4.0.1 [62]. We found that bee species richness, flower species richness, MAT and MAP showed spatial auto-correlation. Bee abundance and flower density did not show spatial auto-correlation (see Results). In the case of the four auto-correlated response variables, we followed procedure as described in Zuur et al. [63], to control for spatial auto-correlation. For each response variable vs elevation, we build up seven different models. We used generalized least squares (GLS) models with five different spatial covariance structures (Spherical, Linear, Ratio, Gaussian, and Exponential type of spatial correlation). Then, for each relationship, we compared these five models and the model with no spatial covariance structure, and the 'null model' (with no explanatory variable and no spatial covariance structure). Then, we selected the best-fit model using second-order Akaike information criterion. Finally, to obtain unbiased parameter estimates, we calculated the selected model with restricted maximum likelihood estimates (REML). GLS analyses were conducted with the R package 'nlme' [64]. Adjusted-pseudo R$^2$ of these GLS models were calculated based on the likelihood-ratio test performed with the 'r.squaredLR' function of R package 'MuMIn' [65]. In the case of the two no auto-correlated response variables (bee abundance and flower density) we run a GLS model with no spatial covariance structure and with REML estimation. For each model, we also obtained and adjusted-pseudo R$^2$ in the same way as described above. In all cases, we checked if models complied with the assumptions of normality and homoscedasticity.

As our bee community had three clearly dominant species (see 'Results' section), we decided decomposing bee abundance variable, considering the three most abundant species abundances separately as variables vs elevation, to check if general pattern could be determined by the specific patterns of these three species. We also considered the variables: 'abundance of bees without 3 most abundant species' and 'abundance of bees without the most abundant bee' vs elevation to check directly the effects of these 3 most abundant species in bee abundance vs elevation model. The statistical procedure to conduct these analyses was exactly equivalent to that described for analyses in the previous paragraph. Analyses were conducted

considering elevation as a continuous and as a categorical variable. Details for these analyses and results for these models are shown in the S6 Appendix.

**Bee community structure vs explanatory variables (flower and climate variables).** Disentangling the effect of elevation is problematic since it is often correlated with several abiotic and biotic environmental variables [66], which is also our case (see S2 Appendix). To deal with this problem, we decided to analyze separately elevation as explanatory variable (analyses in previous paragraph), and all those abiotic and biotic variables to which is related, in our case: MAT, MAP, flower species richness and flower density. To test if these four explanatory variables associated to elevational gradient influenced bee community structure response variables i.e., bee species richness and bee abundance, for each one of these two bee response variables we built a series of 'lm' models with all possible combinations of the four explanatory variables (including a 'null model' with no explanatory variables), and then we selected the best-supported models using second-order Akaike information criterion (AICc). This approach reduces the problems associated with multiple testing, collinearity of explanatory variables, and small sample sizes [67]. The best supported models were selected based on their AICc weights, which reveal the relative likelihood of a given model—based on the data and the fit—scaled to one. Model selection was carried out using the 'dredge' function in the MuMIn package for R. The relevant variables were those that were retained in the best-supported models (except, obviously, when the best-supported model consisted only of the intercept). We selected those models with a delta (AICc difference) of $\Delta<2$ and then proceed to run a model-average effect sizes for the parameters with most support across these models with the 'model. avg' function of MuMIn package for R. Model averaging consists in making inference based on a subset of best candidate models, instead of basing conclusions on a single 'best' model [68]. To control for spatial auto-correlation, for each of the two response variables (bee species richness and bee abundance) set of models, we selected the best-supported model based on AICc weight (containing only significant explanatory variables resulting from the model-averaging procedure) and then proceed as described in Zuur et al. [63], as explained in the above paragraph.

**Community composition vs elevation analyses.** To assess the significance of community dissimilarity along the elevation gradient, we used PERMANOVA as implemented in the 'adonis2' function in the 'vegan' package [69] for R. We used 'adonis2' function (instead of 'adonis') to run marginal tests in order to obtain variable effects considering the presence of all other variables in the model (i.e., effects controlling for all other variables). We used NMDS ordinations to visualize community dissimilarities vs elevation. In this specific case we use elevation as a categorical variable for visualization purposes (we defined three elevational categories: 'low', 'mid', and 'high' categories, as described in 'Variable Elevation' section). Ordination plots were created using the 'metaMDS' function in 'vegan' package, which incorporated a square root transformation and Wisconsin double-standardization of species abundances. In both NMDS and PERMANOVA computation we used abundance-based metrics (Bray–Curtis dissimilarity index). We also used PERMANOVA analyses to quantify the contribution of climatic variables (MAT and MAP), and flower variables (flower species richness and flower density). Similarly as explained in above paragraph, we analyzed separately elevation as explanatory variable, from all those abiotic and biotic variables to which is correlated (MAT, MAP, flower species richness and flower density, see S2 Appendix). So, we run a first analysis with community composition as response variable vs elevation as unique explanatory variable, and then a second analysis with community composition as response variable vs climatic & flower variables as explanatory variables. To take into account the effects of undersampling and rare species on community dissimilarity, we compared our NMDS and PERMANOVA results to those generated with all singletons removed, and all singletons and doubletons removed, and

to results calculated with presence-absence distance metrics (Jaccard dissimilarity index). To explore patterns of spatial autocorrelation, we used the 'ecodist' package [70] in R to do Mantel and partial Mantel tests. Community composition showed correlation with geographic distance (see Results), so we tested the significance of elevation and climatic and flower variables on communities while statistically constraining the variation attributable to distance alone. To do so, we used two approaches. First, we used a partial Mantel test to test if there was a relationship between elevation and bee community composition once the effects of geographic distance are removed. Second, we also used the procedure described in Zimmerman and Vitousek [71], conducting a series of constrained distance-based redundancy analysis ('dbRDA') implemented with 'dbRDA' function ('vegan' package) using principal components of neighbor matrices (PCNM). We obtained seven vectors from applying PCNM procedure with 'pcnm' function ('vegan' package). Then we selected significant PCNM vectors with 'capscale' and 'ordistep' functions ('vegan' package), using a both forward and backward selection method. Only significant PCNM vectors were used in dbRDA analyses. We run a first dbRDA analysis including community composition distances as response variable vs elevation + significant PCNM vectors as explanatory variables. Then, we run a second dbRDA analysis including community composition distances as response variable vs PCNM significant vectors + climatic variables (MAT and MAP) + flower variables (flower species richness and flower density) as explanatory variables. We used marginal anova testing in order to obtain results controlling for all other variables in the model (i.e. we obtain effects of explanatory variables after controlling for spatial effects (PCNM vectors)). We conducted these dbRDA analyses for our principal analysis using abundance-based dissimilarities (Bray-Curtis), and also for secondary comparative analyses removing singletons, singletons+doubletons, and for analysis using presence/absence-based dissimilarities.

Finally, we also conducted all analyses described above considering variable Elevation as a categorical variable, for comparison. As results were qualitatively equivalent they are not shown in main text, but they are shown in S5 Appendix.

## Results

### General results

A total of 1726 specimens of bees were captured and sorted into 62 species and morphospecies, included in five families: Apidae (27 species/morphospecies), Megachilidae (7), Andrenidae (9), Halictidae (18) and Colletidae (1) (S3 Appendix). Sixteen species/morphospecies represented 95.08% of the specimens captured, and 23 of the remaining 46 species/morphospecies were singletons. *Macrotera sp1* was the most abundant species (29.3% of total specimens), followed by *Lasioglossum (Dialictus) sp1* (18.6%), and *Lasioglossum (Lasioglossum) sp1* (16.4%). These three species together constitute almost two thirds of the collected individuals. Plot species richness ranged between 6 and 21, and abundance between 18 and 151 (considering 3 surveys merged together). The relationship between bee species richness and bee abundance failed significance (Pearson r = -0.20, p = 0.4). Of the total 1726 specimens, 226 individuals corresponded to *Apis mellifera*, and 1500 to 'solitary bees'. Only solitary bees were used for analyses (as explained in 'Materials & Methods').

### Bee community structure, flower and climate variables vs elevation

Bee species richness significantly decreased with elevation (Table 1, Fig 2A), as did MAT (Table 1, Fig 2C). Richness of flower species significantly increased with elevation (Table 1, Fig 2E). Bee abundance and flower density tended to increase with increasing elevation but failed significance in both cases (Table 1, Fig 2B–2F respectively). MAP showed a "U"-pattern with

**Table 1. Best GLS models for different response variables vs elevation.**

| GLS model parameters | | | | |
|---|---|---|---|---|
| **GLS Model** | | **F** | **p-value** | **pseudo-$R^2$** |
| **Bee species richness ~ Elevation** | | **15.44** | **0.0011** | **0.48** |
| Bee abundance[*1] ~ Elevation | | 1.57 | 0.226 | 0.08 |
| MAT ~ Elevation | | 806.14 | **<0.0001** | 0.98 |
| MAP ~ Elevation + Elevation$^2$ | Elevation | 45.40 | **<0.0001** | 0.85 |
| | Elevation$^2$ | 43.27 | **<0.0001** | |
| Flower species richness ~ Elevation | | 17.13 | **0.0007** | 0.51 |
| Flower density ~ Elevation | | 1.87 | 0.19 | 0.10 |

Best GLS models for different response variables vs elevation (considered as continuous variable), after controlling for spatial autocorrelation. These results are plotted in Fig 2. Significant values are marked in bold. Abbreviations: MAT: Mean Annual Temperature (˚C); MAP: Mean Annual Precipitation (mm). ([*1]: log10-transformed)

significant lower values at mid-elevation and significant higher values at higher elevation (Table 1, Fig 2D). These results were not affected by spatial auto-correlation as, in all cases, models including no spatial correlation structure were selected (had the lower values of AICc).

## Bee community structure vs flower and climate explanatory variables

The model including only temperature as explanatory variable was the best ranked model based in AICc weight for bee species richness as response variable. Model-averaging estimation of parameters confirmed this result, as temperature was the only explanatory variable that resulted significant (see Table 2 and Fig 3). In the case of bee abundance as response variable, the model including only flower abundance as explanatory variable was the best ranked model based in AICc weight. This was confirmed by the model-averaging estimation of parameters, as flower abundance was the only explanatory variable that resulted significant (see Table 2 and Fig 3). Both analyses sets (bee species richness and bee abundance), yielded almost identical results for full average and conditional average, so only conditional average is shown (Table 2). We selected the two best models in each case (Bee species richness ~ temperature, and Bee abundance ~ flower abundance) and checked them for spatial correlation. Results were not affected by spatial auto-correlation as, in two cases, models including no spatial correlation structure were selected (had the lower values of AICc). Finally, parameters of these GLS best models after controlling for spatial autocorrelation were calculated (see Table 2).

## Community composition vs elevation and flower and climate explanatory variables

Community composition dissimilarity showed correlation with increasing geographic distance (Mantel test: r = 0.428, p = 0.002). However, community composition dissimilarities still showed positive correlation with increasing elevation once the effects of geographic distance are removed (partial Mantel test: Rm = 0.23, p = 0.02).

We found significant community composition dissimilarities along elevation (PERMANOVA: F = 4.99, p = 0.002, $R^2$ = 0.23; Table 3A). These community differences along elevation can be visualized in NMDS ordination plot (Fig 4). Elevation effect still was significant after controlling for geographic distance effect (dbRDA: Elevation: F = 2.15, p = 0.044, $R^2$ = 0.08; Table 3A), and geographic distance also resulted significant in explaining differences in community composition (dbRDA: pcnm1: F = 2.34, p = 0.045, $R^2$ = 0.09; pcnm6: F = 2.52, p = 0.031, $R^2$ = 0.10; Table 3A). In analyses considering climatic and flower explanatory

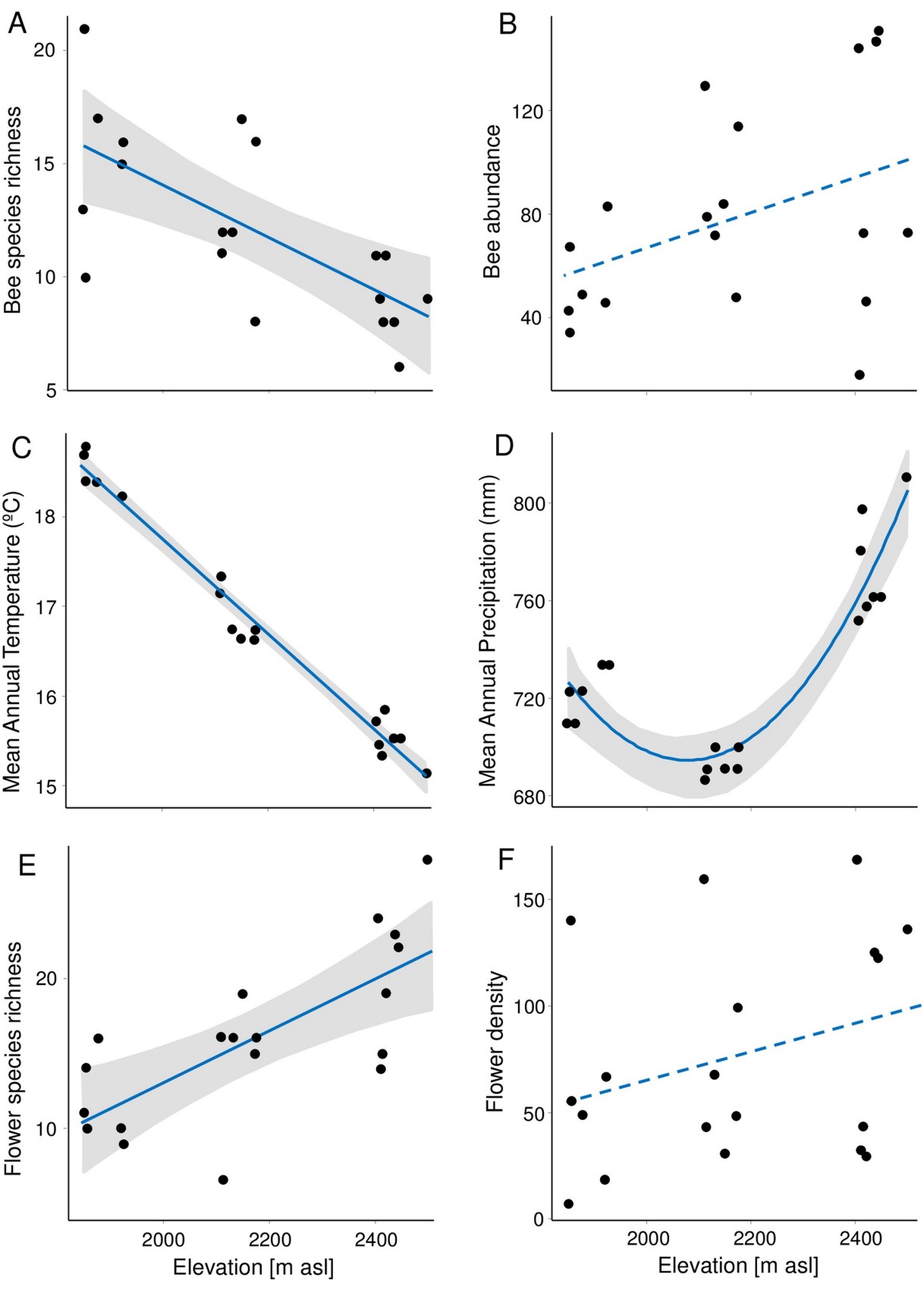

**Fig 2. Effects of elevation.** Effects of elevation (as a continuous variable, in meters above sea level, [m asl]), on different response variables: (A) bee species richness (number of bee species); (B) bee abundance (number of bee individuals); (C) Mean Annual Temperature (MAT,˚C); (D) Mean Annual Precipitation (MAP, mm); (E) flower species richness (number of flower species); (F) flower density (number of flowers/m2); along elevation. Continuous blue lines represents best adjust of significant models, and the gray bands represent 95% confidence intervals. Blue broken lines without gray band denote non-significant models. See in Table 1 the statistic parameters of these models.

variables, MAT and flower density significantly explained a part of the variation in community composition dissimilarities (PERMANOVA: MAT: F = 3.63, p = 0.003, $R^2$ = 0.13; flower density: F = 3.38, p = 0.009, $R^2$ = 0.12; Table 3B), and this effect still remained significant after controlling for geographic distance effect, which also explained a part of variation (dbRDA: MAT: F = 2.75, p = 0.024, $R^2$ = 0.09; flower density: F = 3.03, p = 0.015, $R^2$ = 0.10; pcnm6: F = 2.30, p = 0.043, $R^2$ = 0.07; Table 3B). These results were qualitatively almost the same comparing them with analyses removing singletons, and singletons and doubletons at once, even after controlling by geographic distance effect (Tables A and B in S4 Appendix, respectively). In analyses with binary composition data, results were similar: elevation explained significantly differences in community composition after controlling by geographic distance, and geographic distance failed significance. Further, considering climatic and flower explanatory variables, only temperature had a significant effect (Table C in S4 Appendix).

## Discussion

Our study shows that bee richness significantly decreases with increasing elevation, which seems to be strongly mediated by the effect of decreasing temperature with elevation. On the other hand, bee abundance follows no significant trend along our elevational gradient, although it shows a significant positive relationship with flower density. Bee community

**Table 2. Results from model selection relating bee response variables vs climatic and flower variables.**

| 2A. Model selection results (only models ΔAICc < 2 are presented) | | | | | | | | | | | |
|---|---|---|---|---|---|---|---|---|---|---|---|
| Response variable: **Bee species richness** | | | | | | Response variable: **Bee abundance** | | | | | |
| Model | Explanatory variables included | Estimate | loglik | AICc | ΔAICc | weight | Model | Explanatory variables included | Estimate | loglik | AICc | ΔAICc | weight |
| 1 | MAT | 2.18 | -46.20 | 100.00 | 0.00 | 0.69 | 1 | Flower density | 0.59 | -88.77 | 185.14 | 0.00 | 0.71 |
| 2 | MAT | 2.81 | -45.37 | 101.60 | 1.58 | 0.31 | 2 | Flower density | 0.55 | -88.05 | 187.0 | 1.82 | 0.29 |
|  | Flower species richness | 0.21 | | | | | | MAT | -5.87 | | | | |
| 2B. Model-averaging of parameters included in best ranked models (conditional average) | | | | | | | | | | | |
| Response variable: **Bee species richness** | | | | | Response variable: **Bee abundance** | | | | |
| Explanatory variable | Estimate | adj. se | z-value | p(>\|z\|) | Explanatory variable | Estimate | adj. se | z-value | p(>\|z\|) |
| MAT | 2.37 | 0.72 | 3.28 | **0.001** | Flower density | 0.58 | 0.14 | 4.18 | **<0.0001** |
| Flower species richness | 0.21 | 0.18 | 1.12 | 0.26 | MAT | -5.88 | 5.66 | 1.04 | 0.29 |
| 2C. Best GLS models (after controlling for spatial autocorrelation) | | | | | | | | |
| Model | F | p | pseudo-adj.R$^2$ | Model | F | p | pseudo-adj.-R$^2$ |
| Bee species richness ~ Mean Annual Temperature | 16.25 | **<0.001** | 0.49 | Bee abundance ~ Flower density | 21.95 | **<0.001** | 0.56 |

Best fitting models relating Bee species richness and Bee abundance (as response variables) vs climatic (MAT, MAP) and flower variables (Flower species richness, Flower density). In Table 2A. best ranked models (ΔAICc < 2) are represented for each of bee response variables (Bee species richness (left) and Bee abundance (right)). In Table 2B. model-averaged parameters for explanatory variables are presented for each bee response variable. In Table 2C. parameters for best (GLS) models (after controlling for spatial correlation) are presented. Abbreviations: MAT: Mean Annual Temperature (˚C); MAP: Mean Annual Precipitation (mm); adj.: adjusted; se: standard error. Significant values are marked in bold.

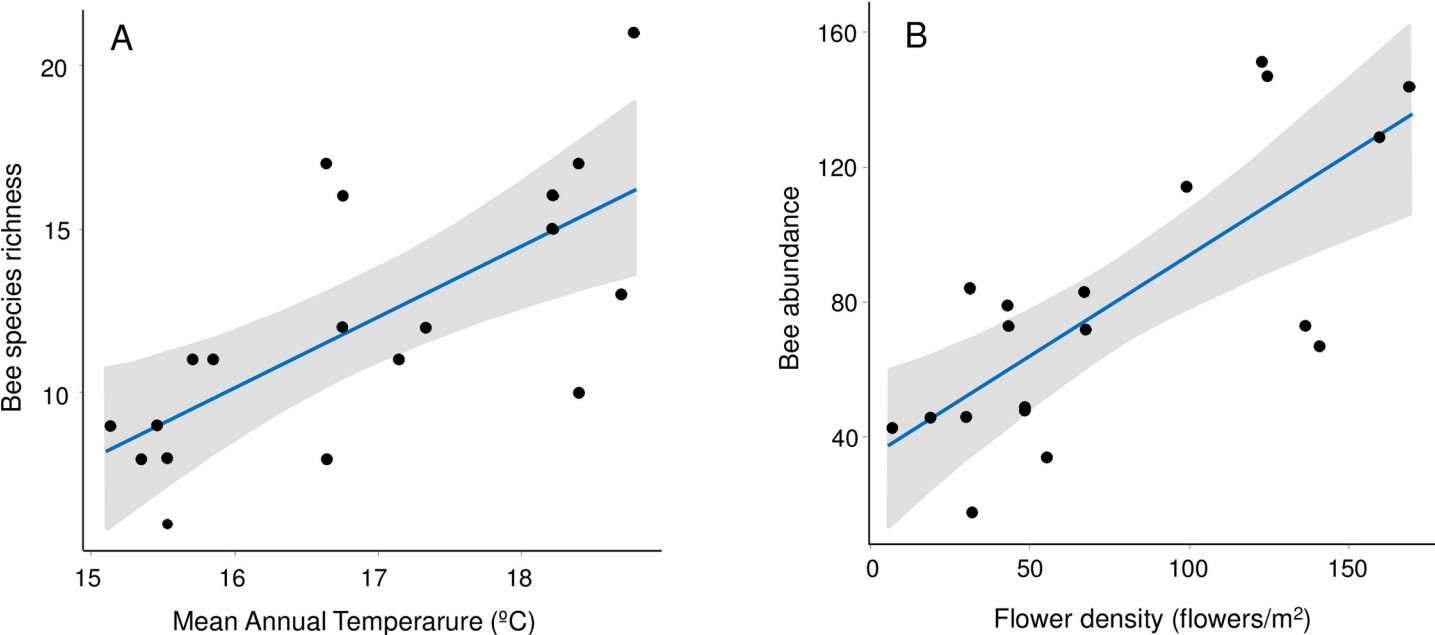

**Fig 3. Best GLS models relating bee response variables vs climatic and flower variables.** (A) Bee species richness vs Mean Annual Temperature (MAT;˚C), and (B) bee abundance vs flower density (number of flowers/m$^2$).

composition shows significant changes with elevation that can be explained by temperature and flower density. These results highlight the importance of climatic conditions on bee community structure and composition [40, 42], and are relevant to our understanding on how bee communities may respond to climate change [23]. In addition, our study represents, as far as we know, the first bee community record for a Mexical-type scrubland (S3 Appendix), which undoubtedly is the least studied ecosystem in Mexico [3, 6]. Some other studies have focused on bee communities in nearby areas, but under different climates and vegetation types and at lower elevations (below 1800 m asl) [11, 72].

## Bee richness and abundance

In our study, as expected, bee species richness showed a significant trend to decrease with increasing elevation, which is in agreement with previous literature describing a pattern commonly observed in insects in general, with either a mid-elevation peak or a monotonical decrease in species diversity over the entire elevational gradient [34, 35, 37, 38, 50]. Specifically for bee species richness, a monotonically decrease with increasing elevation is mostly reported [39–47]. We also found that mean annual temperature was the best predictor for bee species richness, as we expected, showing a clear positive relationship. This is in agreement with other studies which also found a clearly stronger effect of temperature, outweighing the effect of resources [40, 42]. Temperature is known to have effects on species richness directly and indirectly (mediated via an influence on abundance; e.g. [42]). First, only a few species are expected to physiologically tolerate the harsh and cold climates of high-elevation or high-latitude habitats, in animals and plants in general [36, 73, 74], and in bees in particular [75, 76]. In this sense, elevation has been described as acting as an environmental filter on bee communities, excluding individuals that are not adapted to stressful mountain conditions (cold temperatures, wind, short growing seasons) [40, 77]. Bees are known to be typically associated with warm and sunny environments and their species richness peaks in arid-temperate climates

**Table 3. Community composition vs elevation, geographical distance, climatic variables and flower variables.**

| 3A. Considering Elevation and geographic distance as explanatory variables | | | | | |
|---|---|---|---|---|---|
| **PERMANOVA** | | | | | |
| variable | Df | Sum of Squares | R$^2$ | F | P(>F) |
| Elevation | 1 | 0.692 | 0.226 | 4.991 | **0.002** |
| Residual | 17 | 2.357 | 0.773 | | |
| Total | 18 | 3.050 | 1 | | |
| **dbRDA (controlling for geographic distance)** | | | | | |
| variable | Df | Sum of Squares | R$^2$ | F | P(>F) |
| pcnm1 | 1 | 0.277 | 0.09 | 2.341 | **0.045** |
| pcnm6 | 1 | 0.299 | 0.10 | 2.526 | **0.031** |
| Elevation | 1 | 0.255 | 0.08 | 2.152 | **0.044** |
| Residual | 15 | 1.77 | 0.58 | | |
| Total | 18 | 3.05 | 1 | | |
| 3B. Considering climatic (temperature and precipitation), flower (flower species richness and flower density), and geographic distance as explanatory variables | | | | | |
| **PERMANOVA** | | | | | |
| variable | Df | Sum of Squares | R$^2$ | F | P(>F) |
| Mean Annual Temperature (˚C) | 1 | 0.4 | 0.13 | 3.63 | **0.003** |
| Flower density (flowers/m$^2$) | 1 | 0.37 | 0.12 | 3.38 | **0.009** |
| Flower species richness | 1 | 0.14 | 0.05 | 1.28 | 0.247 |
| Mean Annual Precipitation (mm) | 1 | 0.16 | 0.05 | 1.47 | 0.168 |
| Residual | 14 | 1.54 | 0.51 | | |
| **dbRDA (controlling for geographic distance)** | | | | | |
| variable | Df | Sum of Squares | R$^2$ | F | P(>F) |
| pcnm1 | 1 | 0.13 | 0.04 | 1.32 | 0.258 |
| pcnm6 | 1 | 0.23 | 0.07 | 2.30 | **0.043** |
| Mean Annual Temperature (˚C) | 1 | 0.27 | 0.09 | 2.75 | **0.024** |
| Flower density (flowers/m$^2$) | 1 | 0.30 | 0.10 | 3.03 | **0.015** |
| Flower species richness | 1 | 0.15 | 0.05 | 1.52 | 0.198 |
| Mean Annual Precipitation (mm) | 1 | 0.03 | 0.01 | 0.26 | 0.972 |
| Residual | 12 | 1.19 | 0.39 | | |

PERMANOVA and dbRDA analyses. In Table 3A., Elevation (in PERMANOVA), or Elevation + geographic distance variables (pcnm1 and pcnm6) (in dbRDA) are considered as unique explanatory variables. Elevation is considered as a continuous variable. In Table 3B., climatic variables (Mean Annual Temperature and Mean Annual Precipitation) and flower variables (flower species richness and flower density) (in PERMANOVA), or climatic and flower variables + geographic distance variables (pcnm1 and pcnm6) (in dbRDA), are considered as explanatory variables. In all cases community composition is response variable, and quantitative matrix and Bray-Curtis dissimilarity index is applied.

[76, 78, 79]. Second, ambient temperature may determine how much of the potential available resources are accessible to ectothermic organisms, as net profit of foraging animals declines with decreasing temperatures [80, 81]. Such limitations could result in shrinking population densities and increasing probabilities of species extinction in cooler climates [42].

In the case of bee abundance, we found no pattern along our elevation gradient, which is in disagreement with our expectations based on several studies that show a decrease or a mid-elevation peak in bee abundance with increasing elevation [39–41, 43, 45, 51, but see 42, 52]. Nevertheless, we also found, as expected, that flower density was the best predictor of bee abundance, showing a clear positive relationship, which is in agreement with previous works along elevational gradients [40, 42, 51, 52], and it is also known from some studies not associated with elevational gradients [53, 54].

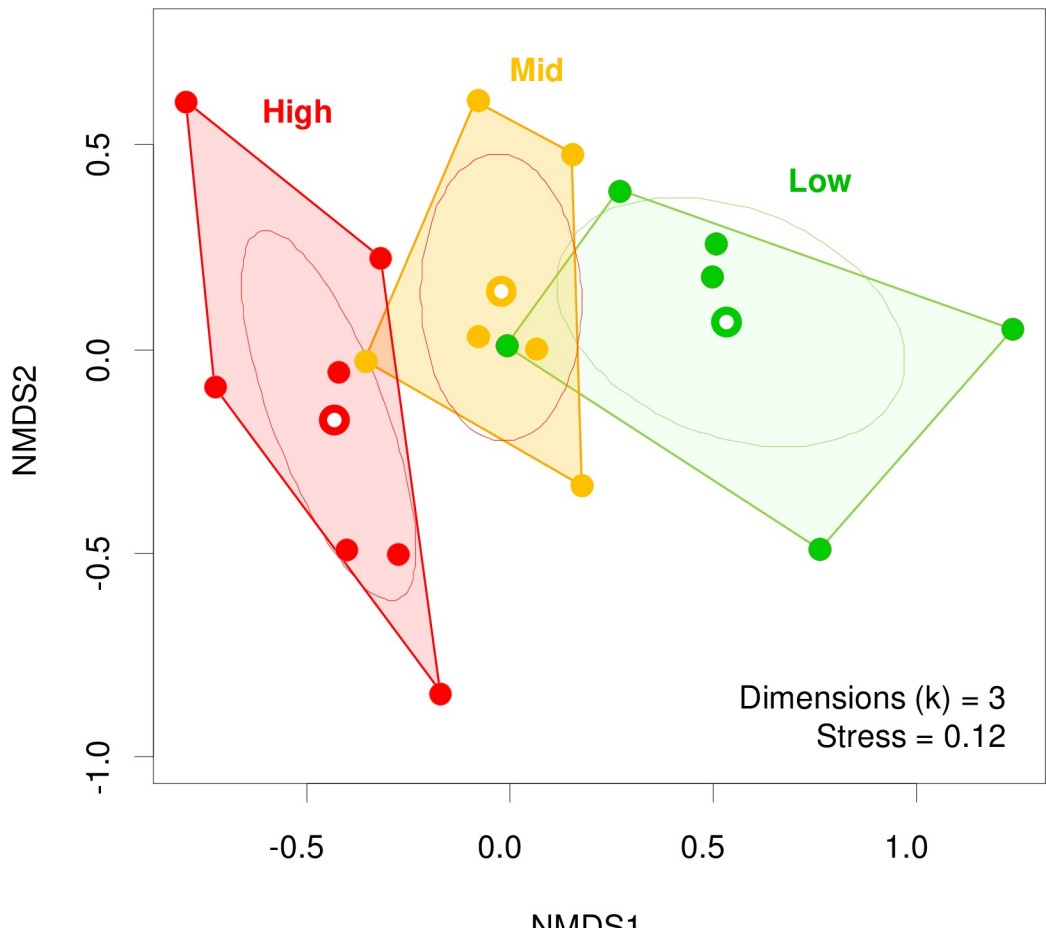

**Fig 4. Community composition vs elevation.** NMDS for quantitative bee community composition (relative abundance of all species in each plot, considering three surveys lumped together). In this analysis we considered Elevation as a categorical variable for visualization purposes. Each dot corresponds to a plot (red dots = 'high' elevation category, orange dots = 'mid' elevation category, green dots = 'low' elevation category). Dots with white center are 'centroids' for each elevation category. Polygons encompass all sites within an elevation category. Ellipses represent 0.95% confidence intervals. Only two of the three dimensions obtained in the analyses (k = 3) are displayed.

Most studies along elevational gradients have found similar trends for bee richness and abundance [39–45, 47, 51]. By contrast, we found different patterns for bee richness and abundance. This result is probably related to a differential effect of climatic conditions on these two variables along our elevational gradient. Our results suggest that temperature is restrictive enough in our elevational gradient to negatively affect bee richness, resulting in a decreasing trend of bee richness with increasing elevations (and lower temperatures). In the case of bee abundance, we found no pattern along the elevational gradient, but a positive relationship with flower density. Flower density may also be negatively affected by low temperature [e.g. 82], but temperature seems to be not restrictive enough in our study, as we found no pattern in flower density along the elevational gradient. Consequently, the lack of pattern in flower density may explain the lack of pattern in bee abundance along our elevational gradient. Due to the low latitude of our study area (~17°N), the climatic conditions, even at the highest sites, are not extreme enough to affect flower density. Another study conducted in a low latitude area (Mt. Kilimanjaro; 2°S) also reports similar levels of flower abundance along an elevational gradient (870 to 4550 m asl), but a clear decreasing trend in bee richness with elevation [42].

By contrast, a study from higher latitudes (Alps; 45˚N; 970 to 2700 m asl) found a clear elevational decrease both in flowering plants abundance and bee richness beginning at mid-elevations (1772 m asl) [51].

Although our bee community has three clearly dominant species, the lack of abundance elevational pattern was not determined by the specific patterns of these three species. The overall lack of pattern was maintained when either the most abundant species (*Macrotera sp1*) or the three most abundant species together, were excluded from the analysis (see S6 Appendix). Interestingly, these three species showed contrasting trends: abundance of *Macrotera sp1* (29.3% of total individuals) increased with elevation, *Lasiglossum (Dialictus) sp1* (18.6%) showed no pattern, and *Lasioglossum (Lasioglossum) sp1* (16%) showed a hump-shaped pattern (see S6 Appendix). The case of *Macrotera sp1* is especially interesting because its pattern is inverse to that found in most of the previous elevational studies [39–41, 43, 45, 51]. This suggests a cold-adaptation of this species which would allow it to exploit high elevation flower resources.

Our results are relevant for the response of the Mexical bee community to a foreseeable scenario of climate change [24], with potentially different impacts on bee richness and abundance. In areas in which the Mexical occupies the top of the mountains, precluding upwards migration, some cold-adapted species might go extinct [8, 31, 83, 84]. However, overall bee richness is expected to increase at higher elevations, as global warming would allow low-elevation species to extend their range towards higher elevations [8, 30–32]. A net increase in bee richness in the higher areas of the gradient could result in greater functional complementarity due to a more diversified pollinator community, which could favor pollination function [85, 86]. Bee abundance, on the other hand, is likely to be indirectly affected by climate change through the effects of climate change on flower abundance. In this vein, contrasting results have been reported, with some plant species increasing their flower production with increasing temperatures, while others reacting conversely [23, 87]. However, in the Mexical scrubland, dominant plant species belong to old lineages (Tertiary) which are C3, evergreen-sclerophyllous species [3, 6]. These physiological traits make these plants particularly vulnerable to suffer significant physiological and structural damages, and even plant mortality, under increasing temperatures and hydric stress [88]. In addition, the Mexical vegetation is highly sensitive to long drought episodes, which are expected to increase in duration and intensity, as documented in Mexico in the last two decades [89–91]. This vulnerability becomes especially relevant where the Mexical occupies the top of the mountains, unable to shift its distributional range upwards tracking more favorable conditions. Therefore, we would expect a decrease in primary productivity and flower abundance in the Mexical community, leading to a lowered pollinator abundance [23]. Consequently, we would expect a decline in pollination function [85, 86]. This trend might be reinforced by a population decline of the cold-adapted most abundant bee species at the high elevations (*Macrotera sp1*).

## Bee community composition

Regarding community composition, we found significant and consistent differences along our elevation gradient, as we expected, in agreement with previous works [45, 46, 48, 49]. Community composition changes along elevation are mainly due to temperature and flower density, and to some extent, also to geographic distance. The fact that we found bee compositional differences along the elevational gradient, suggest certain degree of specialization in species ecological niche in terms of elevation and temperature. We found two species abundantly represented (at least 30 individuals at high elevation) that were clearly more abundant at high elevations than in mid or low elevations (*Macrotera sp1*, *Pseudopanurgus sp1*). We also found one species that was exclusive from high and mid-elevations (*Lasioglossum (Dialictus) sp2*). On

the other hand, we found some species that were clearly more abundant (*Halictus sp1b)* or exclusively represented (*Ceratina sp3*) in low elevations. As this elevational species distribution is in great part conditioned by ambient temperature, we can expect that increasing temperatures due to climate change [24] will determine shifts of species' distributions along the elevational gradient, with a gradual replacement of cold-adapted species by warm-adapted ones [8, 31, 32, 92, 93]. In our bee community, those species clearly more abundant at high elevations (*Macrotera sp1*, *Pseudopanurgus sp1*, *Lasioglossum (Dialictus) sp2*), which are expected to have some kind of cold-environment adaptations or preferences, are likely to face physiological affectations or life-cycle mismatches, which could reduce their populations or even go extinct as their habitat progressively shrinks [8, 23, 31, 83, 84]. Meanwhile, we also would expect that those species occupying at present mostly lower to mid-elevation as, for instance, *Lasioglossum (Lasioglossum) sp 1*, *Ceratina* sp3, *C. sp4b*, *Eucerini sp3*, *E. sp6*, or *Halictus sp1b*, with apparent warmer-environment preferences, would be able to expand their distributional ranges upwards [8, 30]. Similar upwards shifts are expected for plant communities along the elevational distribution, associated to climate change [92–95]. If flowering plants and pollinators react to temperature changes at different rates, this could provoke plant-pollinator distributional mismatches [30, 96]. Nevertheless, pollination systems tend towards generalization [97]. If most of bee and plants species are generalist, they may be able to use different partners across their range, so diverging spatial ranges will not necessarily have any fitness impacts on either group [96], and impacts on pollination function could be relatively low. But, it is likely that specialist species would be the most affected in this context [98], and novel communities could be impoverished by the loss of some of these species.

## Concluding remarks

In our system, bee richness is affected mostly by temperature. It is likely that expected increased temperatures under climate change will enhance bee richness in the highest parts of the mountains, as bee species from lower and mid-elevations will tend to move upwards [8, 31, 32]. Nevertheless, extinction of cold-adapted bee species is also possible [8, 31, 83, 84]. On the other hand, we found bee abundance to be affected mostly by resource availability (flower abundance), which, in turn, is dependent on climatic conditions. Increasing temperatures and prolonged drought episodes, are likely to have a strong negative impact on the Mexical scrubland, especially the mountain top areas. This would lead to a decline in flower abundance which, in turn, would result in decreased bee abundance. Although difficult to predict, changes in both flower and bee composition are also expected, potentially altering plant-pollinator interactions. Climate change is also likely to affect the phenology and distributional range of plant and pollinators, potentially leading to temporal and/or spatial mismatches if these two groups of organisms respond differently to weather variables [96], which would further enhance changes in plant-pollinator interactions and potentially threaten pollination function, especially in specialized plant-pollinator systems. Future research could be focused on understand how these changes will be taking place, what community variables will result affected first and in what relative extent, and how it will affect pollination function.

## Supporting information

**S1 Appendix. General database.**
(XLSX)

**S2 Appendix. Correlations among all response and explanatory variables.**
(DOCX)

**S3 Appendix. List of species and/or morphospecies.**
(DOCX)

**S4 Appendix. Mantel test, PERMANOVA and dbRDA analyses for quantitative data excluding sigletons.** excluding sigletons and doubletons, and for binary data (considering elevation as a continuous variable).
(DOCX)

**S5 Appendix. Results of all analyses considering elevation as a categorical variable.**
(DOCX)

**S6 Appendix. Abundance of the three most abundant species in our Mexical community vs elevation, and abundance of bees excluding most abundant species vs elevation.**
(DOCX)

# Acknowledgments

We are most grateful to 'Comisariado de Bienes Comunales' and Municipal Authorities of Villa de Tamazulápam del Progreso, Villa de Chilapa de Díaz, and San Andrés Lagunas (including delegated authorities of San Isidro Lagunas, belonging to San Andrés Lagunas municipality), and in general, to all people of these villages for their welcome and kindness. We are also grateful to all people of San Isidro Lagunas, and especially to Mrs. Angélica Hernández and family, who kindly hosted us and offer us their hospitality and delicious dishes. We are also thankful to Germán Estocapan, Ana Edith Alcántara, Katia Haydeé Torres, Nicolás Osorno, Noé Guzmán, Mario Flores, Benajmín Cruz, Josué Palma and Alan Flores, for their friendship and field assistance. Finally, we are also grateful to the editor and two anonymous reviewers, who contributed with their comments and suggestions to improve this manuscript.

# Author Contributions

**Conceptualization:** Sergio Osorio-Canadas, Alfonso Valiente-Banuet.

**Data curation:** Sergio Osorio-Canadas.

**Formal analysis:** Sergio Osorio-Canadas.

**Funding acquisition:** Sergio Osorio-Canadas, Alfonso Valiente-Banuet.

**Investigation:** Sergio Osorio-Canadas.

**Methodology:** Sergio Osorio-Canadas, Noé Flores-Hernández, Tania Sánchez-Ortiz, Alfonso Valiente-Banuet.

**Project administration:** Sergio Osorio-Canadas, Alfonso Valiente-Banuet.

**Resources:** Alfonso Valiente-Banuet.

**Supervision:** Alfonso Valiente-Banuet.

**Validation:** Alfonso Valiente-Banuet.

**Writing – original draft:** Sergio Osorio-Canadas.

**Writing – review & editing:** Sergio Osorio-Canadas, Noé Flores-Hernández, Tania Sánchez-Ortiz, Alfonso Valiente-Banuet.

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
