## [Decision Letter · Decision Letter 0]

7 Jan 2021

PONE-D-20-34943

Changes in the structure and composition of the ‘Mexical’ scrubland bee community along an altitudinal gradient

PLOS ONE

Dear Dr. Sergio Osorio-Canadas,

Thank you for submitting your manuscript to PLOS ONE. We have received two detailed reviews of your manuscript. Based on their evaluation and my own reading, I am afraid that I cannot recommend publication of your work in its current form. While both reviewers and I think that your study is relevant for the readership of PlosOne there are substantial issues that you need to address before we can consider publication. Reviewer #2 provides excellent comments here I will strongly suggest to carefully follow. Therefore, we invite you to submit a revised version of the manuscript that addresses the points raised during the review process.

We look forward to receiving your revised manuscript.

Kind regards,

François Rigal

Academic Editor

PLOS ONE

Journal Requirements:

2. We note that Figure 1 in your submission contain map images which may be copyrighted. All PLOS content is published under the Creative Commons Attribution License (CC BY 4.0), which means that the manuscript, images, and Supporting Information files will be freely available online, and any third party is permitted to access, download, copy, distribute, and use these materials in any way, even commercially, with proper attribution. For these reasons, we cannot publish previously copyrighted maps or satellite images created using proprietary data, such as Google software (Google Maps, Street View, and Earth). For more information, see our copyright guidelines: http://journals.plos.org/plosone/s/licenses-and-copyright.

2.1.    You may seek permission from the original copyright holder of Figure 1 to publish the content specifically under the CC BY 4.0 license. 

2.2.    If you are unable to obtain permission from the original copyright holder to publish these figures under the CC BY 4.0 license or if the copyright holder’s requirements are incompatible with the CC BY 4.0 license, please either i) remove the figure or ii) supply a replacement figure that complies with the CC BY 4.0 license. Please check copyright information on all replacement figures and update the figure caption with source information. If applicable, please specify in the figure caption text when a figure is similar but not identical to the original image and is therefore for illustrative purposes only.

Reviewers' comments:

Reviewer's Responses to Questions

**Comments to the Author**

1. Is the manuscript technically sound, and do the data support the conclusions?

Reviewer #1: Yes

Reviewer #2: Yes

2. Has the statistical analysis been performed appropriately and rigorously? 

Reviewer #1: Yes

Reviewer #2: Yes

3. Have the authors made all data underlying the findings in their manuscript fully available?

Reviewer #1: Yes

Reviewer #2: Yes

4. Is the manuscript presented in an intelligible fashion and written in standard English?

Reviewer #1: Yes

Reviewer #2: Yes

5. Review Comments to the Author

Reviewer #1: Overall: The authors did a good job summarizing and documenting bee species abundance and richness along their Mexican elevation gradient. Further I like the additional at looking at correlated variables such as temperature, precipitation and floral availability. I had just a few minor comments

Line 29: You don’t talk about this in your paper so you can probably just remove this sentence

Line 45 & line 423: You make the statement throughout your paper that the area you sampled is one of the most diverse places globally. However, this is not the case for bee. Given that you paper is largely about bee diversity I find this statement odd. Can you restructure this claim in your manuscript? See Orr et 2020 Current Biology they talk about the distribution bee richness.

Line 101: generally, you see bees more abundant in lower elevations and flies more abundant in high elevations. Here you claim that bees and flies are equally abundant in low elevations however the three papers you cite, Arroyo, Lefebvre and McCabe all show low abundance of flies at lower elevations. Please correct this sentence in your manuscript

Methods: Can you add a sentence or two in your methods about where you deposited your pinned specimens and who ID these? It also looks like most of your specimens were only IDed down to morphospeices. Is this the lowest taxonomic resolution that you could get them down too?

Do you say what year you did this sampling? I don’t think I saw it in there?

Line 130: Why not put this in km since its such a high value in m?

Line 159: Were these plots all on the same side of the mountain? If not do you think this influence the variation between sites?

Line 168: was this an average between 1902 – 2011 or were you able to extract yearly data based on your sampling time? Either way please specify

Line 171: Was this the time frame that you would have expected to see the most bee activity? Not being from this area I am not sure what kind of bee activity you have during this sampling periods

Line 172: How did you have 3 stations in two rows? Was there an uneven sampling design or was this 2 rows with 3 stations in each row? Please specify

Line 194: Did this remove all your replications? Did you only combine your “seasons”? Could you instead do a mixed model where “season” was a random variable and keep your replications? I only think this is needed if you have an indication that you would get significance in abundance if you increased your samples size.

Line 318: Did you include singletons in your NMDS and PerMANOVA?

Lines 319: Have you thought about documenting the abundance differences for these two species? How do they change along the gradient? Is this driving your abundance trend? I think this should be discussed more

Line 433- 437: I think that the fact that you didn’t find changes in abundance is really interesting and should be discussed more. What do you think is causing this constant? Do you have really common generalist species that occur in all of your sites?

Line 456: Consider adding McCabe et al 2019 “Environmental filtering of body size and darker coloration in pollinator communities indicate thermal restrictions on bees, but not flies, at high elevations” PeerJ They show that in high elevation communities that size and darkness (largely due to temperature) is contributing to the change in pollinator communities along elevation gradients.

Line 482- 483: This sentence is confusing please reword.

Reviewer #2: Review of the manuscript entitled “Changes in the structure and composition of the ‘Mexical’ scrubland bee community along an altitudinal gradient” for PLOS ONE (PONE-D-20-34943).

Comments to the author(s)

In this manuscript the authors sampled bees in different elevations in mountains covered by the Mexical scrubland in Mexico to investigate their patterns of community structure and composition. They found that bee species richness declined with increasing elevation and that this relationship is mediated by temperature, while bee abundance did not follow any pattern, but was positively related to flower density. The bee species composition was influenced by elevation, temperature and flower density. In my opinion, the study is well designed, and the manuscript is well written. However, I have some suggestions that I think that need to be addressed to improve the manuscript quality.

Broad suggestions:

My first major suggestion is that I think the authors could focus their Discussion more on the main findings of their study. I thought it very interesting the contrasting results of bee species richness and abundance because although the lack of pattern against elevation for abundance is rare, as it was related to resources (flower density), it matches ecological theory. I was expecting a deeper discussion of these two patterns in the light of ecological theory and implications for the conservation of ecosystem processes and services. For example, the alterations in temperature will affect the bee species composition and richness, but it will be the alterations in plant communities (and consequently the alterations in flowers) that will most affect the quantity of functions and services that the bees would deliver (mediated by bee abundance). I think the authors could improve this part of discussion, linking it to the paragraph that starts in line 490.

My second suggestion is about the choice of analysing elevation separated in three categories. I understand the rationale and I am aware that the elevations of each plot form three groups but still, there is some variation. So, I think it would be possible to run linear regressions instead of comparing groups (in the same way the authors did for temperature, which is highly correlated to elevation, and there was enough variation). I do not think the way the authors did the analyses is wrong, but quite the opposite, because the statistical analyses were very well conducted, and I congratulate the authors for that. But I think it would be possible to improve the representation of the patterns by considering elevation as a continuous variable.

A third suggestion is to consider changing altitude to elevation throughout the text. Myself used to use altitude when talking about the elevation above sea level, until I understood that the correct form in English is elevation. Altitude is more suitable to meters above land, for example when in a plane.

Minor suggestions:

Line 31 – Change to “…highlight that altitude gradient negatively affects bee species…”

Line 32 – Bee abundance had no… with no comma

Lines 49 to 54 – This sentence is too long. Consider dividing into two.

Lines 59 to 60 – Develop more the ideas. How are the origins and ecology of the Mexical different from the lowlands? Explain it.

Line 61 – Change to “As the Mexical occupies relatively high elevations in mountain systems…”

Line 62 to 63 – The mexical is repeating here. Maybe change to “strong impact on this ecosystem.”

Line 82 – I think the authors could improve the link between this paragraph and the previous. I thought it was too abrupt.

Line 84 – On the other hand is repeated here. The previous sentence also started with “on the other hand”.

Line 85 – Insert a comma: “…populations will decline, potentially leading…”

Lines 95 to 102 – I thought this paragraph is out of the logical flow of the introduction, especially the part between lines 98 and 102. Consider rephrasing it.

Line 106 – Cut the s in “changes”.

Lines 108 to 109 - I think the end of this paragraph could be improved a lot. I would expect at least some hypothesis and prediction for example for objectives 2 and 3. The author could also explain what they were expecting to find that would help to “provide insights to predict how climate change may affect pollinator communities”.

Line 130 – Delete the word apart.

Line 176 – Brilliant could be replaced by shiny?

Line 190 – Change to “…totally comparable, which was our main concern.”

Line 197 to 198 – Was this species present in all plots? Include this information here.

Lines 207 to 209 – Change the verbs to the past.

Line 237 – There is something strange in this sentence: “…estimation, and obtained and adjusted-pseudoR2 in the…”

Line 238 – Change to “In all cases, we checked if models complied…”

Line 331 – Remove the word And before MAP.

Figures – All the figures seem to be with low resolution. I do not know if it was just because it is a first version for review, but I think it worth to look at it.

Figure 2 and 3 – I suggest removing the grey background to make a cleaner version of the graphs.

Lines 354 and 357 – The word resulted could be changed to remained?

Lines 396 to 399 – I did not understand this sentence. If geographic distance failed significance, why would it need to be controlled firstly?

Lines 399 to 401 – The way the sentence is written is strange. “Flower density failed significance explaining community composition.” In addition, what about climatic variables? Were they significant?

Lines 422 to 426 – I think the authors could start their discussion with the implications of the most important results. Although relevant, the fact that this was the first study on bees in the Mexical is not the most important part of this manuscript. The authors can use these sentences in the end of the first paragraph of discussion, but I suggest starting it with a general view of the implications of their results.

Line 427 – Change tendency to trend.

Lines 428 to 429 – I do not think the authors could say there is a trend while it is not statistically significant.

Line 434 – Explore better the contrast between your findings on the abundance patterns and the literature. In this sentence you just say your results disagree with other studies, but how and why? I think this whole paragraph could be improved with the suggestion I made above about discussing the general patterns and the contrast between richness and abundance.

Line 464 – Change find to found.

Lines 467 to 469 – This is one of the most intriguing result, so the authors could explore more the reasons why it happens.

Line 472 – Change work to works.

Line 485 to 489 – This inference is too abrupt. The authors should develop more these ideas. That climate change can be important in determining changes in the altitudinal distribution is something that we already know, but the authors have evidence to discuss how it will happen? Which ecosystem processes and functions will be impaired?

Lines 522 to 524 – I agree that functional ecology will play an important role in elucidating these patterns, but how this study contributed to this? For example, in this manuscript the authors found that richness is affected by temperature, but abundance is affected by resource availability. Future research, for example, should focus on understand which community parameter will be affected first and consequently will affect ecosystem processes. I suggest developing more these ideas based on what the authors found.

6. PLOS authors have the option to publish the peer review history of their article (what does this mean?). If published, this will include your full peer review and any attached files.

Reviewer #1: No

Reviewer #2: No

---

## [Author Response · Author response to Decision Letter 0]

31 Mar 2021

Response to Reviewers

Reviewer #1: 

Overall: The authors did a good job summarizing and documenting bee species abundance and richness along their Mexican elevation gradient. Further I like the additional at looking at correlated variables such as temperature, precipitation and floral availability. I had just a few minor comments

Authors: Thank you, we appreciate your comment.

Line 29: You don’t talk about this in your paper so you can probably just remove this sentence

Authors: We have removed this sentence in the new version of manuscript.

Line 45 & line 423: You make the statement throughout your paper that the area you sampled is one of the most diverse places globally. However, this is not the case for bee. Given that your paper is largely about bee diversity I find this statement odd. Can you restructure this claim in your manuscript? See Orr et 2020 Current Biology they talk about the distribution bee richness.

Authors: We have deleted these sentences referring to Mediterranean richness because they were referred specifically to vegetation species, and we agree with the referee that our main concern are bees in this paper.

Line 101: generally, you see bees more abundant in lower elevations and flies more abundant in high elevations. Here you claim that bees and flies are equally abundant in low elevations however the three papers you cite, Arroyo, Lefebvre and McCabe all show low abundance of flies at lower elevations. Please correct this sentence in your manuscript.

Authors: We agree. Nevertheless, we have rewritten this final paragraph of Introduction as a Referee #2 suggestion, and we have deleted the sentence to which this suggestion of Referee #1 is referred to.

Methods: Can you add a sentence or two in your methods about where you deposited your pinned specimens and who ID these? It also looks like most of your specimens were only IDed down to morphospecies. Is this the lowest taxonomic resolution that you could get them down too?

Authors: We have added a sentence in the new version of the manuscript (see lines 197-198). All bees collected were curated and identified to genus-level by the authors in the Ecology Institute of Autonomous National University of Mexico (UNAM). Bees were identified authors using the ‘Bee Genera IDnature guide’ from DiscoverLife.org (Ascher and Pickering 2011) and ‘The Bee Genera of North and Central America’ (Michener et al. 1994). For most part of genera in our study area, there are no taxonomical revisions published. So, for species-level identification we would have needed to visit a reference collection from a locality relatively close to ours (Zapotitlán Salinas Valley, Puebla; ~130 km far away from our study zone), which is deposited at the Entomological Collection, Universidad de las Americas-Puebla, Cholula, Puebla, Mexico (Vergara & Ayala 2002). This collection has a total of 3487 specimens corresponding to 259 species. Although the Zapotitlán Salinas Valley is located in a more arid region and with lower altitudes than our study area, its relative proximity allows us to think that it may contain a good representation of the regional fauna and many of the species in this area may be common to our study area. Unfortunately, restrictions on mobility due to the covid-19 pandemic did not allow us to visit this collection, so we were forced to present bee identifications to morphospecies-level in our manuscript.

Do you say what year you did this sampling? I don’t think I saw it in there?

Authors: Yes, this information is mentioned in the first sentence of ‘Bee sampling’ paragraph (“We conducted 3 surveys (late September 2019, late October 2019, and late January 2020-early February 2020”) (see lines 176-177).

Line 130: Why not put this in km since it’s such a high value in m?

Authors: Done (line 133).

Line 159: Were these plots all on the same side of the mountain? If not do you think this influence the variation between sites?

Authors: Yes, all the plots were on the same side of the mountain. We carefully took this point into account in the plot selection process. We had pointed out this in the manuscript, as follows: ‘We choose plots with similar conditions of slope and aspect as far as possible’. (See lines 162-163).

Line 168: was this an average between 1902 – 2011 or were you able to extract yearly data based on your sampling time? Either way please specify

Authors: Data obtained from raster layers for GIS of ‘Climatic Atlas of Mexico’ were one value per plot corresponding to an average between 1902 and 2011. We did not obtain a value for our sampling time. We have rephrased this sentence to make this clearer, as follows: ‘We obtained these two variables from corresponding raster layers for GIS of ‘Climatic Atlas of Mexico’ [49]. We obtained Mean Annual Temperature (ºC) (MAT, henceforth) and Mean Annual Precipitation (mm) (MAP, henceforth) for each plot. These data represent an average from a series of 109 years recorded data for during years 1902 to 2011.’ (See lines 171-174).

Line 171: Was this the time frame that you would have expected to see the most bee activity? Not being from this area I am not sure what kind of bee activity you have during this sampling periods.

Authors: There is no study about bee phenology in our study zone. However, it exists a study about bee phenology in Zapotitlán Salinas (Vergara & Ayala 2002), a locality not far away to our study zone (~ 130 Km far), although with lower elevation and a more arid climate. Even considering this aridity differences, our study area climogram exhibit two peaks of precipitation in summer (June and September), which is a very similar pattern to that found in Zapotitlán Salinas climogram. In our latitudes, rain is the main trigger of flowering events, which are likely tracked by main events of pollinators abundance. As in Zapotitlán Salinas’ study most of the bee genera showed a very clear abundance peak in September, we assumed that a similar pattern could be expected in our study zone.

Line 172: How did you have 3 stations in two rows? Was there an uneven sampling design or was this 2 rows with 3 stations in each row? Please specify

Authors: We had 2 rows with 3 stations in each row (for a total of 6 stations per plot and survey). We have added a short phrase in the new version of the manuscript to clarify this point, as follows: ‘In each survey and plot, we placed 6 sampling stations distributed in two parallel rows (3 stations in each of the two rows)’. (See lines 177-178). 

Line 194: Did this remove all your replications? Did you only combine your “seasons”? Could you instead do a mixed model where “season” was a random variable and keep your replications? I only think this is needed if you have an indication that you would get significance in abundance if you increased your samples size.

Authors: We only lumped our “seasons”, which did not were our replications. Our replications were the plots (n=19, 6 for ‘low-elevation category, 6 for ‘mid’ and 7 for ‘high’). Although we had no clear indication that we would get significance in bee abundance increasing sample size (keeping seasons without lumping them together), we conducted the analysis that reviewer suggested us. The result was similar, bee abundance failed significance (Linear Mixed Model with ‘Elevation’ as a continuous variable: F1,53=0.46, p=0.5; Linear Mixed Model with ‘Elevation’ as a categorical variable: F2,52=0.72, p=0.49; in both cases after controlling for spatial autocorrelation).

Line 318: Did you include singletons in your NMDS and PerMANOVA?

Authors: We conducted four versions of NMDS, PerMANOVA and dbRDA analyses: with all individuals (including singletons), without singletons, without singletons and doubletons, and a qualitative version (with presence/absence matrix). Results were almost identical (only qualitative version showed some little differences). We show analysis with all individuals (including singletons) in the main manuscript text, and we included the three remaining versions as ‘Supporting information’ (S4 Appendix, S5 Appendix, S6 Appendix). This is described in the lines 291-294 and 315-318 of ‘Material and Methods’.

Lines 319: Have you thought about documenting the abundance differences for these two species? How do they change along the gradient? Is this driving your abundance trend? I think this should be discussed more

Authors: In the new version of the manuscript, at this point of community description, we have included a third most abundance species (Lasioglossum (Lasioglossum) sp1) that was not mentioned in the first version of the manuscript. Now this sentence is as follows: ‘Macrotera sp1 was the most abundant species (29.3% of total specimens), followed by Lasioglossum (Dialictus) sp1 (18.6%), and Lasioglossum (Lasioglossum) sp1 (16.4%). These three species together constitute almost two thirds of the collected individuals’. (See lines 329-331).

With regard to the suggestion of the reviewer, we have conducted some exploratory analyses to take a look in the abundance of each one of this three most abundant species along the elevational gradient. We found that each species showed a different trend: Macrotera sp1 tend to increase its abundance towards higher altitudes (Linear model (LM) with Elevation as a categorical variable: F=10.8, p=0.001, R2=0.52; LM with Elevation as continuous variable: F=15.33, p=0.0011, R2=0.44, in both cases abundance log10-transformed); Lasioglossum (Dialictus) sp1 showed no trend (LM with Elevation as categorical variable: F=0.35, p=0.71, R2=-0.07; LM with Elevation as continuous variable: F=0.028, p=0.87, R2=-0.057), and Lasioglossum (Lasioglossum) sp1 showed a hump-shaped trend (LM with Elevation as categorical variable: F=3.68, p=0.048, R2=0.23; LM with Elevation as continuous variable: Elevation: F=0.02, p=0.88, Elevation2: F=10.14, p=0.006, R2=0.31, in both cases abundance log10-transformed). 

(Caution: see plots in the word document attached "Response to Reviewers")

These three trends combined could, effectively, determine the lack of a clear trend in the global analysis ‘Bee abundance vs Elevation’. However, although these three most abundant species could be contributing to mask any possible pattern, it seems that they are not determinant in the global pattern found, as the lack of pattern remains even when we exclude the most abundant species (Macrotera sp1: LM with Elevation as categorical variable: F=1.65, p=0.22, R2=0.067; LM with Elevation as continuous variable: Elevation: F=0.8, p=0.38; Elevation2: F=2.85, p=0.11; R2=0.084); or the three most abundant species together from analyses (LM with Elevation as categorical variable: F=0.82, p=0.46, R2=-0.02; LM with Elevation as continuous variable: F=1.66, p=0.21, R2=0.035, in both cases abundance log10-transformed).

We have included a paragraph at the ‘Discusion’ in the new version of the manuscript, making reference to these results (see lines 473-482), and we also have included a new document of Supporting Information with the detailed results and plots (see S7 Appendix).

Line 433- 437: I think that the fact that you didn’t find changes in abundance is really interesting and should be discussed more. What do you think is causing this constant? Do you have really common generalist species that occur in all of your sites?

Authors: We agree with the reviewer. Thanks. We have included a paragraph in the new version of the manuscript discussing this point (see lines 483-499). We think that, as bee abundance is positively correlated with flower abundance, the fact that we found no pattern in flower abundance along the elevational gradient may explain the lack of pattern in bee abundance along our elevational gradient. We also think that, the lack of pattern in flower abundance in our study is due to the low latitude of our study area (~17ºN), which make that the climatic conditions, even at the highest sites, are not extreme enough to affect flower density. Similar results have been described in previous low-latitudes studies (Classen et al. 2015).

Unfortunately, we have no information about generalization/specialization habits of the bee species we found, so we are not able to respond the last question.

Line 456: Consider adding McCabe et al 2019 “Environmental filtering of body size and darker coloration in pollinator communities indicate thermal restrictions on bees, but not flies, at high elevations” PeerJ They show that in high elevation communities that size and darkness (largely due to temperature) is contributing to the change in pollinator communities along elevation gradients.

Authors: We have added this new reference (line 459). Thanks.

Line 482- 483: This sentence is confusing please reword.

Authors: We have rewritten this sentence in the new version of manuscript (see lines 531-532).

--------------

Reviewer #2: Review of the manuscript entitled “Changes in the structure and composition of the ‘Mexical’ scrubland bee community along an altitudinal gradient” for PLOS ONE (PONE-D-20-34943).

Comments to the author(s):

In this manuscript the authors sampled bees in different elevations in mountains covered by the Mexical scrubland in Mexico to investigate their patterns of community structure and composition. They found that bee species richness declined with increasing elevation and that this relationship is mediated by temperature, while bee abundance did not follow any pattern, but was positively related to flower density. The bee species composition was influenced by elevation, temperature and flower density. In my opinion, the study is well designed, and the manuscript is well written. However, I have some suggestions that I think that need to be addressed to improve the manuscript quality.

Authors: Thank you for your comments. We appreciate them.

Broad suggestions:

My first major suggestion is that I think the authors could focus their Discussion more on the main findings of their study. I thought it very interesting the contrasting results of bee species richness and abundance because although the lack of pattern against elevation for abundance is rare, as it was related to resources (flower density), it matches ecological theory. I was expecting a deeper discussion of these two patterns in the light of ecological theory and implications for the conservation of ecosystem processes and services. For example, the alterations in temperature will affect the bee species composition and richness, but it will be the alterations in plant communities (and consequently the alterations in flowers) that will most affect the quantity of functions and services that the bees would deliver (mediated by bee abundance). I think the authors could improve this part of discussion, linking it to the paragraph that starts in line 490.

Authors: We agree with the reviewer, thanks. We have included a paragraph in the new version of the manuscript where we discuss a possible explanation (see lines 483-522 in the new version of the manuscript). 

We think that this result is probably related to a differential effect of climatic conditions on bee richness and abundance along our elevational gradient. Our results suggest that temperature is restrictive enough in our elevational gradient to negatively affect bee richness, resulting in a decreasing trend of bee richness with increasing elevations (and lower temperatures). In the case of bee abundance, we found no pattern along the elevational gradient, but a positive relationship with flower density. Flower density may also be negatively affected by low temperature, but temperature seems to be not restrictive enough in our study for plants, as we found no pattern in flower density along the elevational gradient. Consequently, the lack of pattern in flower density may explain the lack of pattern in bee abundance along our elevational gradient. We think that this could be related to the low latitude of our study area (~17ºN), determining that the climatic conditions, even at the highest sites, are not extreme enough to affect flower density. Similar results have been described in previous low-latitudes studies (Classen et al. 2015).

We also discuss how climate change, and specially increasing temperatures, may determine alterations in flower abundance which, in turn, may determine possible effects for bee abundance and pollination services.

My second suggestion is about the choice of analysing elevation separated in three categories. I understand the rationale and I am aware that the elevations of each plot form three groups but still, there is some variation. So, I think it would be possible to run linear regressions instead of comparing groups (in the same way the authors did for temperature, which is highly correlated to elevation, and there was enough variation). I do not think the way the authors did the analyses is wrong, but quite the opposite, because the statistical analyses were very well conducted, and I congratulate the authors for that. But I think it would be possible to improve the representation of the patterns by considering elevation as a continuous variable.

Authors: We run linear regressions considering elevation as a continuous variable as the reviewer suggested, but the results were almost identically to those comparing groups (we display the results and plots in the next pages). We also conducted multivariate analyses (PERMANOVA and dbRDA) using elevation as a continuous variable. Again results were very similar, in all cases elevation resulted significant (and also one of the ‘pcnm’ vectors representing geographic distance now resulted significant), but amount of variability (R2) explained by elevation was lower than using elevation as categorical variable. As a disadvantage of using Elevation as a continuous variable, in the NMDS plot, it is not possible to represent groups of elevation, so this visualization of results would lose clarity. We consider that, taking into account that results were qualitatively equivalent using elevation as a continuous variable, we would prefer to maintain analyses comparing groups of elevation, as the study was designed for it (but we have added these results using Elevation as a continuous variable as a Supplementary Information, see S8 Figures and Tables). However, if the editor and the reviewer consider that it is preferable presenting analyses with elevation as a continuous variable, we have no inconvenient in changing it.

At the moment, we have shown this results (using Elevation as a continuous variable) as a Supplementary material (see S8 Appendix), and we have made reference to that in the main text of the new version of the manuscript (see lines 164-166 and 319-321). 

(Caution: see Fig.2 and tables in attached word document "Response to Reviewers)

New results for linear model analyses:

Table 1. Best GLS models for different response variables vs Elevation (as a continuous variable).

GLS model parameters

GLS Model F p-value pseudo-R2

Bee species richness ~ Elevation 15.44 0.0011 0.48

Bee abundance* ~ Elevation 1.57 0.226 0.08

MAT ~ Elevation 806.14 <0.0001 0.98

MAP ~ Elevation + Elevation2 Elevation 45.40 <0.0001 0.85

 Elevation2 43.27 <0.0001 

Flower species richness ~ Elevation 17.13 0.0007 0.51

Flower density ~ Elevation 1.87 0.19 0.10

(*log10-transformed)

New results for multivariate analyses (Community composition vs Elevation):

Table 3. Community composition vs Elevation (as a continuous variable), geographical distance, climatic variables and flower variables. (Complete quantitative matrix)

3.A. Considering Elevation as explanatory variable (continuous)

PERMANOVA 

variable Df Sum of Squares R2 F P(>F)

Elevation 1 0.692 0.226 4.991 0.002

Residual 17 2.357 0.773 

Total 18 3.050 1 

dbRDA (controlling for geographic distance) 

variable Df Sum of Squares R2 F P(>F)

pcnm1 1 0.277 0.09 2.341 0.045

pcnm6 1 0.299 0.10 2.526 0.031

Elevation 1 0.255 0.08 2.152 0.044

Residual 15 1.77 0.58 

Total 18 3.05 1 

New results for multivariate Analyses included as Supporting Information in the former version of manuscript:

S4 Table. Community composition vs Elevation, geographical distance, climatic variables and flower variables. (Quantitative matrix excluding singletons).

S4.A. Considering Elevation as explanatory variable (continuous)

PERMANOVA 

variable Df Sum of Squares R2 F P(>F)

Elevation 1 0.696 0.233 5.173 0.001

Residual 17 2.289 0.766 

Total 18 2.985 1 

dbRDA (controlling for geographic distance) 

variable Df Sum of Squares R2 F P(>F)

pcnm1 1 0.273 0.091 2.391 0.036

pcnm6 1 0.298 0.100 2.619 0.030

Elevation 1 0.250 0.084 2.197 0.047

Residual 15 1.711 0.573 

Total 18 2.985 1 

S5 Table. Community composition vs Elevation, geographical distance, climatic variables and flower variables. (Quantitative matrix excluding singletons and doubletons)

S5.A. Considering Elevation as explanatory variable (continuous)

PERMANOVA 

variable Df Sum of Squares R2 F P(>F)

Elevation 1 0.688 0.235 5.244 0.002

Residual 17 2.233 0.764 

Total 18 2.922 1 

dbRDA (controlling for geographic distance) 

variable Df Sum of Squares R2 F P(>F)

pcnm1 1 0.265 0.090 2.405 0.044

pcnm6 1 0.305 0.104 2.765 0.028

Elevation 1 0.244 0.083 2.212 0.040

Residual 15 1.656 0.566 

Total 18 2.922 1 

S6 Table. Community composition vs Elevation, geographical distance, climatic variables and flower variables. (Qualitative (binary) matrix)

S6.A. Considering Elevation as explanatory variable (continuous)

PERMANOVA 

variable Df Sum of Squares R2 F P(>F)

Elevation 1 0.654 0.153 3.087 0.001

Residual 17 3.601 0.846 

Total 18 4.255 1 

dbRDA (controlling for geographic distance) 

variable Df Sum of Squares R2 F P(>F)

pcnm1 1 0.264 0.062 1.304 0.171

pcnm5 1 0.298 0.070 1.468 0.079

Elevation 1 0.338 0.079 1.668 0.034

Residual 15 3.044 0.715 

Total 18 4.255 1 

A third suggestion is to consider changing altitude to elevation throughout the text. Myself used to use altitude when talking about the elevation above sea level, until I understood that the correct form in English is elevation. Altitude is more suitable to meters above land, for example when in a plane.

Authors: We have changed ‘altitude’ to ‘elevation’ throughout the manuscript.

Minor suggestions:

Line 31 – Change to “…highlight that altitude gradient negatively affects bee species…”

Authors: Done (lines 29-30).

Line 32 – Bee abundance had no… with no comma

Authors: Done (line 31).

Lines 49 to 54 – This sentence is too long. Consider dividing into two.

Authors: Done (line 48).

Lines 59 to 60 – Develop more the ideas. How are the origins and ecology of the Mexical different from the lowlands? Explain it. 

Authors: Mexical has an evolutionary origin and ecological traits that are much closer to other Mediterranean ecosystems, than to the low deciduous forest and derived srublands, located a few meters below in the same mountain. In fact, Mexical biome represents the same vegetation associated to Mediterranean climates, sharing common plant genera, and common traits (such as evergreen-esclerophyllous leaves, leaf angle, and the ability to resprout). So, Mexical scrubland could be considered a relict of the evergreen-sclerophylous Mediterranean-like vegetation that originally existed in a belt around North America and Eurasia during the mid-Eocene, from which the other five Mediterrean-type ecosystems existing nowadays also evolved [Axelrod 1958, 1975]. During the transition from warm and wet Tertiary to dry Quaternary, Mediterranean climate appeared and surviving taxa seek refuge in today’s current Mediterranean areas, whereas Mexical, which did not experienced this climatic transition, refuged along the principal mountain chains of Mexico, between ~1850 to 2500 meters above sea level (m asl) (Valiente-Banuet & Verdú 2020). This would explain all the ecological traits and plant genera that the Mexical shares with the species of these other Mediterranean areas (Valiente-Banuet & Verdú 2020). Contrarily, deciduous forest and derived srublands (spiny scrublands, for instance), located at lower strata of the mountains (around 1800 m a.s.l. and below), belong to a different botanical lineage (neotropical geoflora), and have different ecological traits (Rzedowski 1978).

We have included a couple of sentences explaining this in the new version of the manuscript (see lines 58-62).

Line 61 – Change to “As the Mexical occupies relatively high elevations in mountain systems…”

Authors: Done (line 63).

Line 62 to 63 – The mexical is repeating here. Maybe change to “strong impact on this ecosystem.”

Authors: Done (line 64-65).

Line 82 – I think the authors could improve the link between this paragraph and the previous. I thought it was too abrupt.

Authors: We agree with the reviewer, thanks. We have modified this link between paragraphs (see line 83).

Line 84 – On the other hand is repeated here. The previous sentence also started with “on the other hand”.

Authors: Done (We have changed ‘On the one hand’ in the first sentence to ‘For instance’) (line 84).

Line 85 – Insert a comma: “…populations will decline, potentially leading…”

Authors: Done (line 87).

Lines 95 to 102 – I thought this paragraph is out of the logical flow of the introduction, especially the part between lines 98 and 102. Consider rephrasing it.

Authors: We have rewritten and relocated this paragraph in the new paragraph at the final part of Introduction, as a part of revised literature in which we base our expectations for objectives 2 and 3 (see lines 103-110).

Line 106 – Cut the s in “changes”.

Authors: Done (line 100).

Lines 108 to 109 - I think the end of this paragraph could be improved a lot. I would expect at least some hypothesis and prediction for example for objectives 2 and 3. The author could also explain what they were expecting to find that would help to “provide insights to predict how climate change may affect pollinator communities”.

Authors: We have rewritten this final paragraph of the Introduction, introducing some expectations for objectives 2 and 3, based on revised literature (see lines 103-110). 

Line 130 – Delete the word apart.

Authors: Done (line 133).

Line 176 – Brilliant could be replaced by shiny?

Authors: We have changed “brilliant” to “bright” following some literature* about pan traps (line 181).

(*see, for instance: ‘The Utility of Aerial Pan-Trapping for Assessing Insect Pollinators Across Vertical Strata’. Nuttman et al. 2011).

Line 190 – Change to “…totally comparable, which was our main concern.”

Authors: Done (line 195).

Line 197 to 198 – Was this species present in all plots? Include this information here.

Authors: Yes, this species was present in all plots. We have included this information in the new version in the manuscript (see line 205).

Lines 207 to 209 – Change the verbs to the past.

Authors: Done (see lines 214 and 215).

Line 237 – There is something strange in this sentence: “…estimation, and obtained and adjusted-pseudoR2 in the…”

Authors: We have rewritten this sentence in the new version of the manuscript (see lines 244-245).

Line 238 – Change to “In all cases, we checked if models complied…”

Authors: Done (line 245).

Line 331 – Remove the word And before MAP.

Authors: Done (line 342).

Figures – All the figures seem to be with low resolution. I do not know if it was just because it is a first version for review, but I think it worth to look at it.

Authors: The figures were checked to accomplish quality standards of the journal, using the software (‘PACE’) provided by the journal itself in its web page. In my screen, the figures look fine, but, may be, they do not have resolution enough. I would like the journal could confirm to me that figures are good enough, please.

Figure 2 and 3 – I suggest removing the grey background to make a cleaner version of the graphs.

Authors: Done.

Lines 354 and 357 – The word resulted could be changed to remained?

Authors: We prefer to maintain the word ‘resulted’, because ‘remained’ seems to make reference to some other variable that in a former analysis was significant, but, after a following analyses was not significant any more, but this is not the case. In our analyses, first we obtained a best model following AICc criteria (but no significance values), and in the second analytical step, we obtained significance of the variables included in the best AICc model (lines 365 and 368 in the new version of the manuscript).

Lines 396 to 399 – I did not understand this sentence. If geographic distance failed significance, why would it need to be controlled firstly?

Authors: In this case, the first analytical step to decide if we need to taking into account geographic distance was a Mantel test (Community composition dissimilarities ~ Geographic distance), which resulted significant. So, geographical distance seems to be playing a role in the composition differences, and we need to taking into account it in the next analytical step, which is the dbRDA (or PERMANOVA), where we consider if there are a relationship between changes in composition and elevation, controlling for geographic distance. When we conduct this second analytical step (dbRDA), considering the model: Composition ~ Elevation + Geographical distance, is when we obtain the result that geographical distance is not significant, when we consider Elevation variable in the same model. It is likely that geographical distance is not significant in these second analyses because Elevation is explaining a more significant part of composition variability.

In the first version of this manuscript we only specify the result of Mantel test for the complete quantitative version of our compositional matrix which was the one described in main text, but we did not give the Mantel test for each of the three other complementary analyses that we gave as supporting information (without singletons, without singletons and doubletons, and with a qualitative compositional matrix). In the new version of the manuscript, we give the result of Mantel test for each one of these complementary analyses (see Supporting information S4 Appendix, S5 Appendix and S6 Appendix).

Lines 399 to 401 – The way the sentence is written is strange. “Flower density failed significance explaining community composition.” In addition, what about climatic variables? Were they significant?

Authors: We have rewritten this sentence (only temperature was significant). See lines 410-411.

Lines 422 to 426 – I think the authors could start their discussion with the implications of the most important results. Although relevant, the fact that this was the first study on bees in the Mexical is not the most important part of this manuscript. The authors can use these sentences in the end of the first paragraph of discussion, but I suggest starting it with a general view of the implications of their results.

Authors: We agree. We have rewritten the first paragraph of the Discussion including a summary of the most important results and relate it to broad implications in climate change scenario. See lines 433-440.

Line 427 – Change tendency to trend.

Authors: Done (line 446).

Lines 428 to 429 – I do not think the authors could say there is a trend while it is not statistically significant.

Authors: We agree. We have deleted this mention of a ‘trend’ (see line 466).

Line 434 – Explore better the contrast between your findings on the abundance patterns and the literature. In this sentence you just say your results disagree with other studies, but how and why? I think this whole paragraph could be improved with the suggestion I made above about discussing the general patterns and the contrast between richness and abundance.

Authors: We have added a new sentence to explain this abundance pattern in relation with literature (see lines 466-468). We discuss the contrast between richness and abundance a little later (see lines 483-499).

Line 464 – Change find to found.

Authors: Done (line 469).

Lines 467 to 469 – This is one of the most intriguing result, so the authors could explore more the reasons why it happens.

Authors: We have rewritten this part of manuscript. Now we discuss this point at lines 469-472 and 483-499.

Line 472 – Change work to works.

Authors: Done (line 525).

Line 485 to 489 – This inference is too abrupt. The authors should develop more these ideas. That climate change can be important in determining changes in the altitudinal distribution is something that we already know, but the authors have evidence to discuss how it will happen? Which ecosystem processes and functions will be impaired?

Authors: We have rewriting this paragraph in the new version on the manuscript (see lines 534-554). We now make reference to how raising temperatures could affect bee communities, especially causing upwards migrations of species that in the present are occupying mid and lower elevations. We also refer to how changes in climatic conditions could affect flowering plants-bee relationships, causing possible mismatches in their relative position along the elevational gradient, which could determine affectations in pollination function.

Lines 522 to 524 – I agree that functional ecology will play an important role in elucidating these patterns, but how this study contributed to this? For example, in this manuscript the authors found that richness is affected by temperature, but abundance is affected by resource availability. Future research, for example, should focus on understand which community parameter will be affected first and consequently will affect ecosystem processes. I suggest developing more these ideas based on what the authors found.

Authors: We agree. We have deleted this reference to functional ecology. Following reviewer suggestions, we have rewriting this final paragraph in the new version of the manuscript, making reference to our bee richness and abundance, and flower abundance results, and how they could be affected by increasing temperatures in a climate change scenario (see lines 556-572).

---

## [Decision Letter · Decision Letter 1]

9 May 2021

PONE-D-20-34943R1

Changes in the structure and composition of the ‘Mexical’ scrubland bee community along an elevational gradient

PLOS ONE

Dear Dr. Sergio Osorio-Canadas,

Thank you for submitting your manuscript to PLOS ONE. After careful consideration, we feel that it has merit but does not fully meet PLOS ONE’s publication criteria yet, although these minor corrections should be very easy to handle. Therefore, we invite you to submit a revised version of the manuscript that addresses the points raised by reviewer #2.

We look forward to receiving your revised manuscript.

Kind regards,

François Rigal

Academic Editor

PLOS ONE

Journal Requirements:

Reviewers' comments:

Reviewer's Responses to Questions

**Comments to the Author**

1. If the authors have adequately addressed your comments raised in a previous round of review and you feel that this manuscript is now acceptable for publication, you may indicate that here to bypass the “Comments to the Author” section, enter your conflict of interest statement in the “Confidential to Editor” section, and submit your "Accept" recommendation.

Reviewer #1: All comments have been addressed

Reviewer #2: All comments have been addressed

2. Is the manuscript technically sound, and do the data support the conclusions?

Reviewer #1: Yes

Reviewer #2: Yes

3. Has the statistical analysis been performed appropriately and rigorously? 

Reviewer #1: Yes

Reviewer #2: Yes

4. Have the authors made all data underlying the findings in their manuscript fully available?

Reviewer #1: Yes

Reviewer #2: Yes

5. Is the manuscript presented in an intelligible fashion and written in standard English?

Reviewer #1: Yes

Reviewer #2: Yes

6. Review Comments to the Author

Reviewer #1: I have no further comments. I believe that the authors have addressed my original comments thoroughly.

Reviewer #2: Review of the manuscript entitled “Changes in the structure and composition of the ‘Mexical’ scrubland bee community along an altitudinal gradient” for PLOS ONE (PONE-D-20-34943R1).

Comments to the author(s)

I congratulate the authors for their effort in addressing the comments and suggestions made by me and the other reviewer. In my opinion, the authors solved most of the issues and explained the remaining problems. I have only very minor suggestions to help polishing the manuscript.

I understood the authors’ explanation for why to maintain the categorical approach in the manuscript. However, in my opinion, the patterns are much clearer to see with the scatter plots and regression curves, specially because we can see the spread of data, which is not possible to visualise with the current plots. The authors could maintain the Figure 4 with the NMDS using the categories just for visualisation purposes and explain that in Methods. Hence, my suggestion is to invert the graphs showed in the manuscript and S8 Appendix, i.e., moving the categorical approach to supplementary material and bringing the continuous approach to the main manuscript. Again, this is just a suggestion and I believe that the decision of presenting the results of continuous or categorical elevation in the main manuscript is up to the authors.

Line 160 – Change to “Hence, our elevation categories…”

Line 245 – Change to “For each model, we also obtained an adjusted-pseudo R²…”

Line 248 – Change ‘explicative’ to ‘explanatory’.

Line 256 – Change to “…response variables i.e., bee species richness and bee abundance, for each one of these two bee…”

Line 452 – Change ‘MAT’ to ‘mean annual temperature’.

Lines 474 to 483 – I think this paragraph could come after the next one (from lines 484 to 500). In addition, I suggest the authors including one or two sentences in Methods’ section to explain why and how they did these analyses.

7. PLOS authors have the option to publish the peer review history of their article (what does this mean?). If published, this will include your full peer review and any attached files.

Reviewer #1: No

Reviewer #2: No

---

## [Author Response · Author response to Decision Letter 1]

17 May 2021

Response to Reviewers

Reviewer #1

Reviewer #1: I have no further comments. I believe that the authors have addressed my original comments thoroughly.

Authors: Thank you for your comments. We appreciate them.

Reviewer #2

Reviewer #2: Review of the manuscript entitled “Changes in the structure and composition of the ‘Mexical’ scrubland bee community along an altitudinal gradient” for PLOS ONE (PONE-D-20-34943R1).

Comments to the author(s)

I congratulate the authors for their effort in addressing the comments and suggestions made by me and the other reviewer. In my opinion, the authors solved most of the issues and explained the remaining problems. I have only very minor suggestions to help polishing the manuscript.

Authors: Thank you for your comments. We appreciate them.

I understood the authors’ explanation for why to maintain the categorical approach in the manuscript. However, in my opinion, the patterns are much clearer to see with the scatter plots and regression curves, specially because we can see the spread of data, which is not possible to visualise with the current plots. The authors could maintain the Figure 4 with the NMDS using the categories just for visualisation purposes and explain that in Methods. Hence, my suggestion is to invert the graphs showed in the manuscript and S8 Appendix, i.e., moving the categorical approach to supplementary material and bringing the continuous approach to the main manuscript. Again, this is just a suggestion and I believe that the decision of presenting the results of continuous or categorical elevation in the main manuscript is up to the authors.

Authors: We have followed the suggestion of the reviewer, and we have included all analyses results and plots considering elevation as continuous variable in the main manuscript, and we have moved all the analyses results and plots considering elevation as categorical variable to supporting information (S5 Appendix in this new version). Further, in the new version, we have reorganized data contained in other Appendixes: we have included all results for multivariate complementary analyses considering elevation as a continuous variable in only one Appendix (information included in S4, S5 and S6 Appendixes in previous version, is now included in the new version in only one Appendix: S4 Appendix). That’s why we have only 6 Appendixes in this new version (instead of 8 Appendixes in the previous version).

Also, as suggested by the reviewer, we have maintained the Figure 4 (NMDS) using variable elevation as three categories for visualization purposes, and we have explained that in ‘Methods’ (see lines 294-297).

Line 160 – Change to “Hence, our elevation categories…”

Authors: This sentence has been rewritten in the new version on the manuscript (see lines 158-172).

Line 245 – Change to “For each model, we also obtained an adjusted-pseudo R²…”

Authors: Done. Thank you. (see lines 249-250).

Line 248 – Change ‘explicative’ to ‘explanatory’.

Authors: Done. Thank you. (see line 262).

Line 256 – Change to “…response variables i.e., bee species richness and bee abundance, for each one of these two bee…”

Authors: Done. Thank you. (see line 270).

Line 452 – Change ‘MAT’ to ‘mean annual temperature’.

Authors: Done. Thank you. (see line 483).

Lines 474 to 483 – I think this paragraph could come after the next one (from lines 484 to 500). In addition, I suggest the authors including one or two sentences in Methods’ section to explain why and how they did these analyses.

Authors: We have relocated the first paragraph (lines 474 to 483 in previous version) after the next one (lines 484 to 500 in previous version), following reviewer’s suggestion (see now lines 523-532 in the new version of the manuscript). 

As the reviewer suggests, we also have included a couple of sentences in Methods’ section to explain why and how we did these analyses (see lines 252-261). Details for these analyses and results for these models are shown in the S6 Appendix.

---

## [Editor Report · Decision Letter 2]

21 Jun 2021

Changes in the structure and composition of the ‘Mexical’ scrubland bee community along an elevational gradient

PONE-D-20-34943R2

Dear Dr. Sergio Osorio-Canadas,

We’re pleased to inform you that your manuscript has been judged scientifically suitable for publication and will be formally accepted for publication once it meets all outstanding technical requirements.

Kind regards,

François Rigal

Academic Editor

PLOS ONE
---

## [Editor Report · Acceptance letter]

23 Jun 2021

PONE-D-20-34943R2 

Changes in the structure and composition of the ‘Mexical’ scrubland bee community along an elevational gradient 

Dear Dr. Osorio-Canadas:

I'm pleased to inform you that your manuscript has been deemed suitable for publication in PLOS ONE. Congratulations! Your manuscript is now with our production department. 

Kind regards, 

on behalf of

Dr. François Rigal 

Academic Editor

PLOS ONE